# Evolutionary principles of modular gene regulation in yeasts

Dawn A Thompson[1][*][†], Sushmita Roy[1,2,†‡a], Michelle Chan[1,3], Mark P Styczynski[1‡b],
Jenna Pfiffner[1], Courtney French[1], Amanda Socha[1], Anne Thielke[1],
Sara Napolitano[1], Paul Muller[1], Manolis Kellis[2], Jay H Konieczka[1], Ilan Wapinski[1‡c],
Aviv Regev[1,4*]

[1]Broad Institute of MIT and Harvard, Cambridge, United States; [2]Computer Science and Artificial Intelligence Laboratory, Massachusetts Institute of Technology, Cambridge, United States; [3]Computational and Systems Biology Program, Massachusetts Institute of Technology, Cambridge, United States; [4]Department of Biology, Howard Hughes Medical Institute, Massachusetts Institute of Technology, Cambridge, United States

**Abstract** Divergence in gene regulation can play a major role in evolution. Here, we used a phylogenetic framework to measure mRNA profiles in 15 yeast species from the phylum *Ascomycota* and reconstruct the evolution of their modular regulatory programs along a time course of growth on glucose over 300 million years. We found that modules have diverged proportionally to phylogenetic distance, with prominent changes in gene regulation accompanying changes in lifestyle and ploidy, especially in carbon metabolism. Paralogs have significantly contributed to regulatory divergence, typically within a very short window from their duplication. Paralogs from a whole genome duplication (WGD) event have a uniquely substantial contribution that extends over a longer span. Similar patterns occur when considering the evolution of the heat shock regulatory program measured in eight of the species, suggesting that these are general evolutionary principles.

*For correspondence: dawnt@broadinstitute.org (DAT); aregev@broadinstitute.org (AR)

†These authors contributed equally to this work

‡Present address: [a]Biostatistics and Medical Informatics, University of Wisconsin, Madison, United States; [b]School of Chemical and Biomolecular Engineering, Georgia Institute of Technology, Atlanta, United States; [c]Department of Systems Biology, Harvard Medical School, Boston, United States

## Introduction

Divergence in the regulation of gene expression has been repeatedly postulated to play a major role in evolution. Examples of regulatory differences between species were described in a wide range of species including bacteria (*McAdams et al., 2004*), fungi (*Gasch et al., 2004*; *Habib et al., 2012*), flies (*Prud'homme et al., 2007*; *Wittkopp et al., 2008*; *Bradley et al., 2010*), and mammals (*Khaitovich et al., 2006*; *Odom et al., 2007*; *Brawand et al., 2011*; *Lindblad-Toh et al., 2011*; *Perry et al., 2012*). However, the mechanisms through which regulatory systems evolve are still only partially understood, and in most cases the adaptive importance of regulatory changes is unknown (*Lynch, 2007*; *Thompson and Regev, 2009*; *Wohlbach et al., 2009*; *Baker et al., 2012*; *Romero et al., 2012*).

In recent years, comparative genomics approaches have allowed us to begin to trace the evolution of gene regulation at different time scales (*Tuch et al., 2008b*; *Weirauch and Hughes, 2010*; *Brawand et al., 2011*; *Lindblad-Toh et al., 2011*; *Romero et al., 2012*), through two major approaches: (1) characterization of *cis*-regulatory elements in orthologous promoter sequences (*Gasch et al., 2004*; *Tanay et al., 2005*; *Bradley et al., 2010*; *Lindblad-Toh et al., 2011*; *Habib et al., 2012*), and (2) comparative analysis of mRNA profiles and protein–DNA interactions measured across organisms (*Tirosh et al., 2006, 2011*; *Borneman et al., 2007*; *Tuch et al., 2008a*; *Schmidt et al., 2010*; *Wapinski et al., 2010*; *Brawand et al., 2011*; *Romero et al., 2012*). While studies relying on *cis*-regulatory sequences are more prevalent, functional studies of comparative gene regulation are beginning to shed light

**eLife digest** The incredible diversity of living creatures belies the fact that their genes are quite similar. In the 1970s Mary-Claire King and Allan Wilson proposed that a process called gene regulation—which determines when, where and how genes are expressed as proteins—is responsible for this diversity. Four decades later, the central role of gene regulation in evolution has been confirmed in a wide range of species including bacteria, fungi, flies and mammals, although the details remain poorly understood. In recent years it has been suggested that the duplication of genes—and sometimes the duplication of whole genomes—has had a crucial influence on the part played by gene regulation in the evolution of many different species.

Ascomycota fungi are uniquely suited to the study of genetics and evolution because of their diversity—they include *C. albicans*, a fungus that is found in the human mouth and gut, and various species of yeast—and because many of their genomes have already been sequenced. Moreover, their genomes are relatively small, which simplifies the task of working out how it has changed over the course of evolution. It is also known that species in this branch of the tree of life diverged before and after an event in which a whole genome was duplicated.

Ascomycota fungi use glucose as a source of carbon in different ways during aerobic growth. Most, including *C. albicans*, are respiratory and rely on oxidative phosphorylation processes to produce energy. However, a small number—including *S. cerevisiae* and *S. pombe*, two types of yeast that are widely used as model organisms—prefer to ferment glucose, even when oxygen is available. Species that favor the latter respiro-fermentative lifestyle have evolved independently at least twice: once after the whole genome duplication event that lead to *S. cerevisiae*, and once when *S. pombe* and the other fission yeasts evolved.

Thompson et al. have measured mRNA profiles in 15 different species of yeast and reconstructed how the regulation of groups of genes (modules) have evolved over a period of more than 300 million years. They found that modules have diverged proportionally to evolutionary time, with prominent changes in gene regulation being associated with changes in lifestyle (especially changes in carbon metabolism) and a whole genome duplication event.

Gene duplication events result in gene paralogs—identical genes at different places in the genome—and these have made significant contributions to the evolution of different forms of gene regulation, especially just after the duplication event. Moreover, the paralogs produced in whole genome duplication events have resulted in bigger changes over longer periods of time. Similar patterns were observed in the regulation of the genes involved in the response to heat shock in eight of the species, which suggests that these are general evolutionary principles.

The changes in gene expression associated with the respiro-fermentative lifestyle may also have implications for our understanding of cancer: healthy cells rely on oxidative phosphorylation to produce energy whereas, similar to yeast cells, most cancerous cells rely on respiro-fermentation. Furthermore, yeast cells and cancer cells both support their rapid growth and proliferation by using glucose for biosynthesis to support cell division, although this process is not fully understood. Normal cells, on the other hand, use glucose primarily for energy and tend not to divide rapidly.

Thompson et al. found that the genes encoding enzymes in two biosynthetic pathways—one that produces the nucleotides necessary for DNA replication, and one that synthesizes glycine—are induced in respiro-fermentative yeasts but repressed in respiratory yeast cells. The fact that similar changes are observed in the same two pathways when normal cells become cancer cells suggests that these pathways have an important role in the development of cancer. The framework developed by Thompson et al. could also be used to explore the evolution of gene regulation in other species and biological processes.

on how regulatory evolution is linked to functional changes. In particular, it has been suggested (*Lynch and Force, 2000*; *Gu et al., 2004, 2005*; *Teichmann and Babu, 2004*; ; *Conant and Wolfe, 2006*; *Tirosh and Barkai, 2007*; *Wapinski et al., 2007b*) that gene duplication can promote regulatory divergence by either neo-functionalization or sub-functionalization of regulatory mechanisms of the two paralogs.

Among eukaryotes, the *Ascomycota* fungi (*Figure 1A*) provide an excellent model to study the evolution of gene regulation (*Tsong et al., 2003, 2006; Ihmels et al., 2005; Tanay et al., 2005; Field et al., 2008; Hogues et al., 2008; Tirosh and Barkai, 2008; Tsankov et al., 2010, 2011; Baker et al., 2012; Habib et al., 2012*). They include the model organisms *Saccharomyces cerevisiae*, *Schizosaccharomyces pombe* and *Candida albicans*, as well as many non-model, genetically-tractable species with sequenced genomes. Species in the phylogeny diverged before and after a whole genome duplication event (*Wolfe and Shields, 1997; Kellis et al., 2004*) (WGD, *Figure 1A*, star, ~150 mya), allowing us to study the consequences of this evolutionary mechanism (*Wolfe and Shields, 1997; Kellis et al., 2004; Wapinski et al., 2007b*).

Comparative genomics of *Ascomycota* has already shed an important light on the evolution of gene expression. For example, studies in yeast showed that while co-expression of genes in modules can be conserved at substantial distances, the associated regulatory mechanisms often diverge, acquiring new regulators and losing ancestral ones, both for sequence-specific transcription factors (*Tsong et al., 2003, 2006; Tanay et al., 2005; Hogues et al., 2008; Lavoie et al., 2010; Baker et al., 2011, 2012*) and for chromatin organization (*Tirosh and Barkai, 2008; Tsankov et al., 2010, 2011*). In some cases, changes in gene expression and related mechanisms are clearly coupled to other adaptive changes in lifestyle (*Ihmels et al., 2005; Field et al., 2008; Tsankov et al., 2010*), whereas in others they may be the result of neutral 'regulatory drift' (*Tsong et al., 2003, 2006; Lavoie et al., 2010; Baker et al., 2012*). Importantly, evolutionary changes in regulators, facilitated by protein modularity, in cooperative binding with other factors and shifts in protein–DNA interactions contribute toward different paths through a 'hybrid' regulatory state for the ancestral regulatory network to be resolved in to generate the diversity of regulatory network structures observed in modern species (*Baker et al., 2011, 2012; Tuch et al., 2008a*)

Despite these early successes, collecting experimental data across species has remained challenging, and hence most experimental studies rely on two to four species (*Tanay et al., 2005; Tirosh et al., 2006; Lelandais et al., 2008; Wittkopp et al., 2008*), with few exceptions (*Schmidt et al., 2010;*

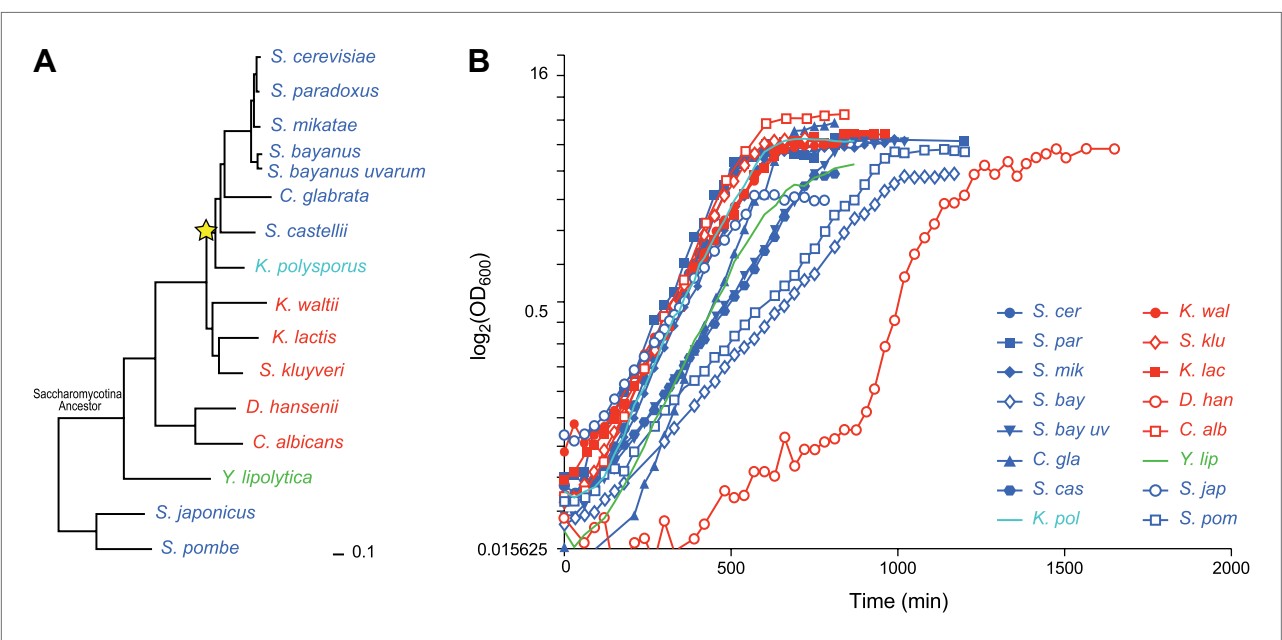

**Figure 1**. *Ascomycota* species in this study. (**A**) A phylogenetic tree of the 15 *Ascomycota* species in the study. Dark blue: respiro-fermentative; red: respiratory; green: obligate respiratory; light blue: intermediate between respiro-fermentative and respiratory. Star: a Whole Genome Duplication event (WGD). (**B**) Growth rate (log(OD)$_{600}$, y axis) of each species over time (y axis) during growth in the novel rich medium used in this study (see 'Materials and methods').

The following source data are available for figure 1:

**Source data 1**. Evolutionary distance across the phylogeny of 15 species.

*Tsankov et al., 2010*; *Wapinski et al., 2010*; *Brawand et al., 2011*; *He et al., 2011*). An important challenge is to collect experimental data in such a way that would minimize irrelevant differences, for example, due to growth conditions, and allow focusing on true evolutionary distinctions. Expanding the experimental scope to cover a broader phylogenetic range and density can help study the divergence of expression in individual genes and gene modules in order to answer questions on the extent of conservation of transcriptional programs, its relation to phylogenetic distance, the emergence of new regulatory patterns across modules of co-regulated genes, and the specific contribution of gene duplication and divergence—through both sporadic and whole genome duplication—to regulatory evolution.

Here, we use comparative transcriptional studies across 15 *Ascomycota* species—spanning >300 million years of evolution (*Sipiczki, 2000*)—to understand the evolution of modular gene regulation during batch growth on glucose and its depletion, a key physiological response. We optimized culture conditions across species, and collected ~300 expression profiles at six physiologically comparable time points along each species' growth: repletion (lag phase), exponential growth ('mid-log' and 'late log'), the point of glucose depletion ('diauxic shift') and two later time points when the growth rate levels off ('post shift' and 'plateau'). To analyze the evolution of regulatory modules, we use a new algorithm, Arboretum (*Roy et al., 2013*), to identify expression modules across species and to reconstruct their evolutionary history.

We find that the degree of divergence of the transcriptional profiles correlates with phylogenetic distance, with the largest divergence in lag phase profiles. In all species, the transcriptional response involves five major transcriptional modules. While the module's expression patterns are conserved across species, their gene membership diverges, proportionally to phylogenetic distance. Gene duplication events significantly contribute to regulatory divergence, in particular close to their phylogenetic point of duplication. This contribution is more pronounced and more prolonged for WGD paralogs. These patterns also characterize the evolution of the transcriptional response to heat shock, supporting their generality (*Roy et al., 2013*). Our framework for comparative functional genomics is applicable to any complex phylogeny, and can help test these principles of regulatory evolution in other responses and species.

## Results

### An experimental system for comparative functional genomics in *Ascomycota*

We studied 15 yeast species whose genome is fully sequenced (*Figure 1A*; 'Materials and methods'; *Table 1* and *Figure 1—source data 1*), spanning >300 million years of evolution (*Sipiczki, 2000*). The species cover the different clades of the phylogeny well, with the exception of the filamentous *Euascomycota*, and have a range of phenotypes related to how they use glucose as a carbon source during aerobic growth. Species in the *Kluyveromyces*, *Candida*, and *Yarrowia* clades are respiratory and use oxidative phosphorylation (*Figure 1A*, red and green). Conversely, a respiro-fermentative lifestyle—a preference to ferment glucose even in the presence of oxygen (*Piskur et al., 2006*)—has evolved independently at least twice in this phylogeny, once after the WGD (*Conant and Wolfe, 2007*) and once in *Schizosaccharomyces* (*Rhind et al., 2011*) (*Figure 1A*, dark blue). *K. polysporus*, the most basal post-WGD species (*Scannell et al., 2007*) has an intermediate phenotype between respiro-fermentative and respiratory (*Figure 1A*, light blue). In contrast to the other respiratory species that can ferment, *Y. lipolytica* (*Figure 1A*, green) is an obligate respiratory species, but can uniquely use normal hydrocarbons and various fats as carbon sources (*Kurtzman, 2000*).

Due to these lifestyle differences some of the species do not grow well in typical media formulations (e.g., YPD). We therefore first optimized our growth medium to minimize growth differences between species ('Media tests' under 'Materials and methods'). Our formulation boosts the growth of otherwise slow growers, without substantially impacting the growth of fast growers (*Figures 1B and 2A*).

### A comparative transcriptional compendium during growth on glucose

Even in our new medium, there is still substantial variation in growth between species, likely indicating real physiological differences, inherent to each species (*Figure 2A*; 'Materials and methods'). We therefore determined in real-time the growth rate, glucose, and ethanol levels for each species ('Materials and methods'; *Figure 2B*, *Figure 2—figure supplement 1*), and chose physiologically comparable (but potentially physically different) time points for each species for isolating RNA from

**Table 1.** Number of genes and orthogroups

| Species | Total genes in species | Total genes on arrays | Total orthogroups on arrays | Total genes with tree*,† | Orthogroups available for analysis‡ | Genes available for analysis§ | Orthogroups analysis 1# | Genes analysis 1# | Orthogroups analysis 2¶ | Genes analysis 2¶ |
|---|---|---|---|---|---|---|---|---|---|---|
| S. cerevisiae | 6343 | 6257 | 4424 | 5508 | 4402 | 5464 | 2746 | 2746 | 3676 | 3964 |
| S. paradoxus | 5512 | 5504 | 4319 | 5256 | 4312 | 5244 | 2577 | 2577 | 3452 | 3720 |
| S. mikatae | 5697 | 5693 | 4251 | 5094 | 4251 | 5093 | 2513 | 2513 | 3382 | 3618 |
| S. bayanus | 5489 | 5483 | 4272 | 5191 | 4269 | 5188 | 2555 | 2555 | 3416 | 3679 |
| C. glabrata | 5338 | 5269 | 4126 | 4909 | 4119 | 4897 | 2534 | 2534 | 3394 | 3614 |
| S. castellii | 5693 | 5689 | 4277 | 5420 | 4257 | 5362 | 2574 | 2574 | 3461 | 3794 |
| K. polysporus | 5328 | 5324 | 4039 | 4539 | 4027 | 4525 | 2506 | 2506 | NA | NA |
| K. waltii | 5198 | 5194 | 4381 | 4849 | 4381 | 4848 | 2560 | 2560 | 3432 | 3497 |
| K. lactis | 5328 | 5323 | 4435 | 4888 | 4428 | 4879 | 2572 | 2572 | 3455 | 3537 |
| S. kluyveri | 5321 | 5320 | 4393 | 4879 | 4386 | 4865 | 2496 | 2496 | 3364 | 3444 |
| D. hansenii | 7938 | 6893 | 4050 | 4635 | 4034 | 4608 | 1903 | 1903 | 2551 | 2634 |
| C. albicans | 6163 | 6107 | 4858 | 5692 | 4858 | 5692 | 2324 | 2324 | 3110 | 3232 |
| Y. lipolytica | 6756 | 6672 | 4260 | 4886 | 4258 | 4874 | 2138 | 2138 | 2855 | 2921 |
| S. japonicus | 5297 | 5149 | 3863 | 4248 | 3861 | 4246 | 1878 | 1878 | 2487 | 2557 |
| S. pombe | 5068 | 5060 | 4208 | 4751 | 4208 | 4750 | 2001 | 2001 | 2487 | 2746 |

Shown are the total number of genes in each species (defined as the sum of genes on arrays and with orthology, 'Materials and methods'). The number of genes, genes that have orthologs in another species, and the classes of genes that were measured on the species-specific arrays (1) total number of genes (2) total number of orthogroups (3) non-singleton (those present *S. cerevisae* and in at least one other species). Also shown is the number of genes and orthgroups resulting after filtering based on a missing value cut of 50% (see 'Materials and methods'). The number of genes and orthogroups per species used in the Arboretum analyses 1 and 2 (without and with duplication).
*Gene trees = orthogroups = orthology.
†This class includes non-singletons that are represented on the microarray.
‡Orthogroups represented on the microarray and satisfy missing values cutoff (50%).
§Genes represented on the microarray and satisfy missing values cutoff (50%).
#Analysis 1: **Figures 5–9** (present in at least one species in addition to *S. cerevisiae*, and did not incur duplication).
¶Analysis 2: **Figures 10–13** (present in at least one species in addition to *S. cerevisiae*, and incurred at most one duplication).
Total number of orthogroups: 7459.

lag, mid-log, late log, 'diauxic shift' (the point at which glucose is depleted), post-shift, and plateau. We compared mRNA levels at each time point to those in a mid-log stage from the same time course in the same species (*Figure 3A*; 'Materials and methods'), thus allowing us to compare differential (relative) expression levels in the response across species. We conducted all experiments in ≥2 biological replicates, which were highly reproducible ('Materials and methods'; *Figure 3—source data 1*). Furthermore, the values measured by arrays were highly consistent with those measured for a selected subset of samples by RNA-Seq, including for duplicated (paralogous) genes ('Materials and methods'). Given this high reproducibility, we present median values across replicates in subsequent analyses, for simplicity.

## Divergence in global expression profiles correlates with phylogenetic distance

To compare the extent of similarity in gene regulation, we calculated Pearson's correlation coefficient between pairs of expression profiles (using median values of biological replicates) for each pair of species at physiologically-matched time points (e.g., post-shift in *S. cerevisiae* and *C. albicans*, 'Materials and methods'). We found that the degree of correlation is inversely related with the phylogenetic distance between the species ($r < -0.39$ to $-0.77$, $p < 3 \times 10^{-5}$, *Figure 3B–F*), such that the closer two species are in the phylogeny, the higher the correlation between their profiles. This inverse relation is apparent at each of the time points (*Figure 3B–F*), and is consistent with what was recently reported in mammals (*Brawand et al., 2011*).

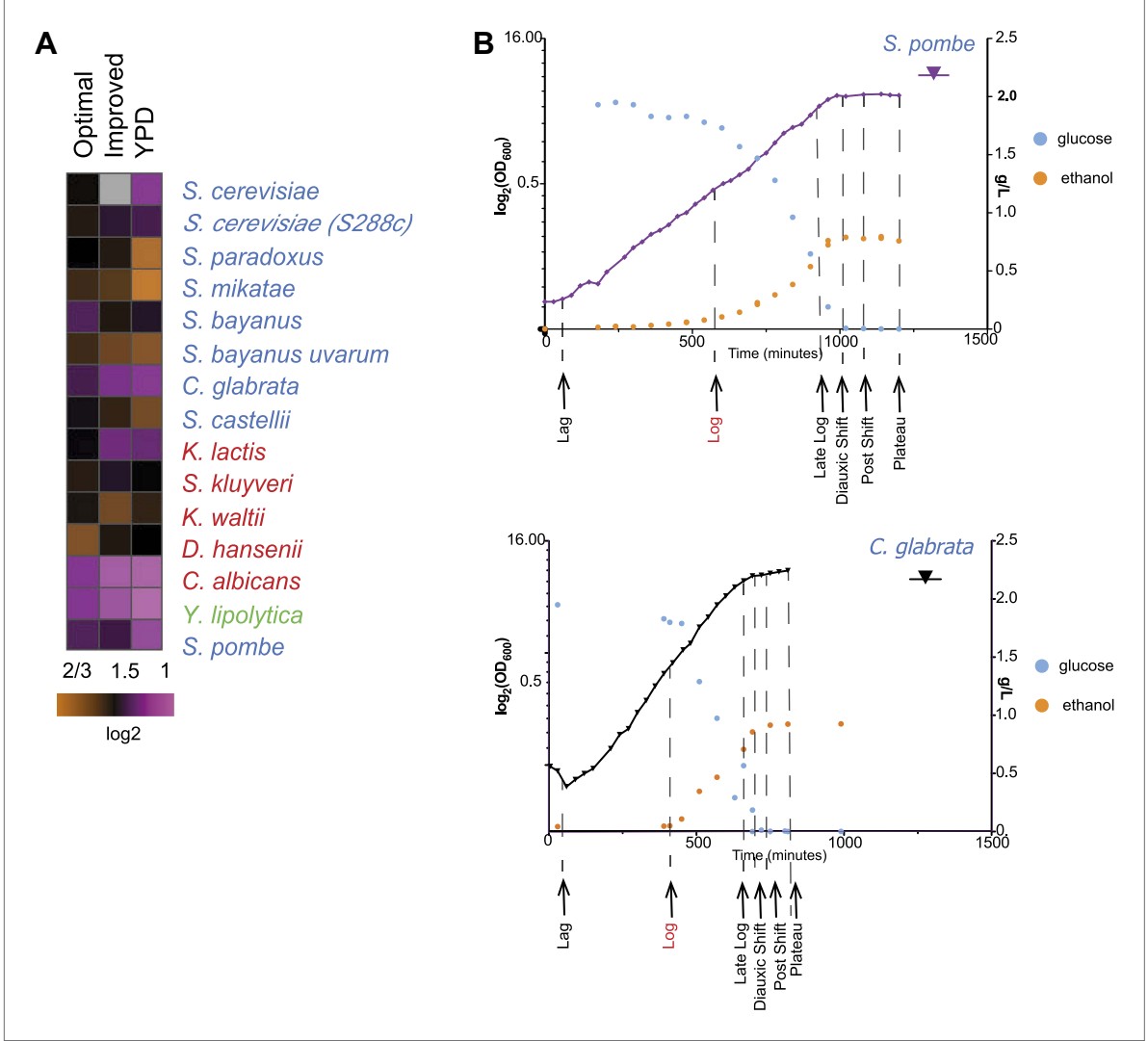

**Figure 2**. Growth of species in published and novel growth media. (**A**) Performance of species in our optimized medium vs YPD medium, a common medium for *S. cerevisiae*. Shown are normalized saturation coefficients (log₂(OD₆₀₀) during a 24-hr growth period, a measure of accumulated biomass) of each species ('Media tests' under 'Materials and methods') in our panel (rows) in three media (columns). (**B**) Choosing 'physiologically comparable' time points. Our experiments compare 'physiologically analogous' time points across all species (see 'Materials and methods'). For example, shown is the growth curve (x axis: time, minutes; y axis: growth rate, in $\log_2(OD_{600})$ and glucose levels (g/L, blue) and ethanol levels (g/L, orange) for the relative slow growing species *S. pombe* (left) vs the growth curve for the faster growing *C. glabrata* (right). Biological samples from each species were taken at the time points indicated by arrows. The Log phase time point (shown in red) used as the reference for microarray analysis.

The following figure supplements are available for figure 2:

**Figure supplement 1**. Phenotypic characterization of each species.

The extent of transcriptional divergence varies between stages. Profiles at diauxic shift, post shift and plateau are the most correlated between species (***Figure 3D–G***), whereas the lag profiles, when cells 'reset' in response to nutrient repletion, are the most divergent (***Figure 3B,G***), likely reflecting species-specific responses to nutrient signals shaped by niche adaptation. *S. japonicus* and *S. castellii* were the most divergent, such that in some cases, their Lag phase profiles were even anti-correlated to those of other species, suggesting distinct repletion programs. In principle, the lower conservation of the lag phase profiles in these and other species could have been merely due to the fact that this is the only phase sampled at the same absolute time (30 min). However, the repletion process is known to be fast (***Zaman et al., 2008***), metabolite profiling in each species at this time point shows the least

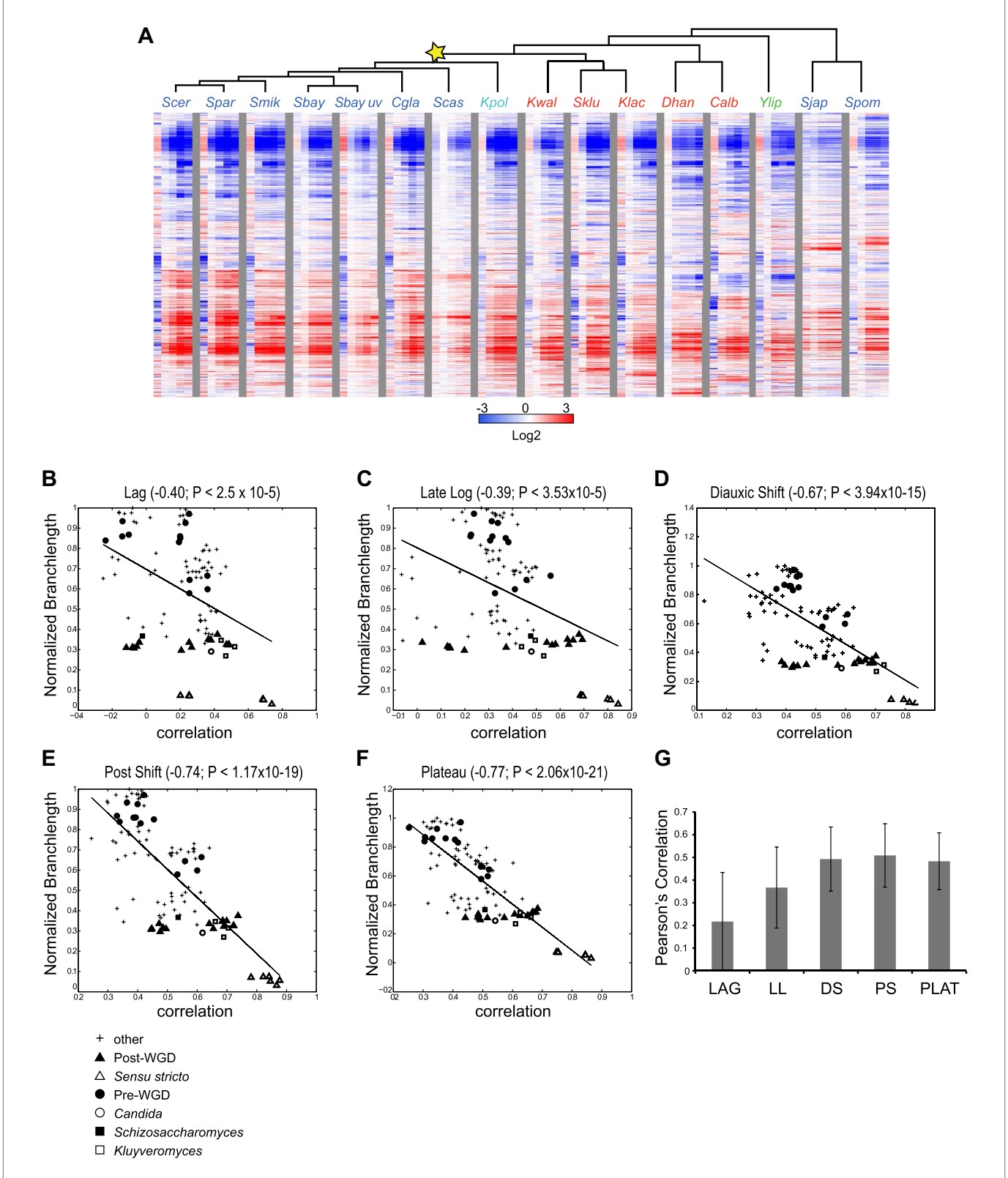

**Figure 3**. Divergence in global expression profiles correlates with phylogenetic distance. (**A**) A comparative transcriptional compendium during growth on glucose. Shown are transcriptional profiles measured for each species (tree, top), at six time points (columns) during growth on glucose: Lag, Late Log, Diauxic Shift, Post Shift and Plateau (left to right). Genes (rows) are matched based on orthology and clustered ('Materials and methods'). Red:

*Figure 4. Continued on next page*

*Figure 3. Continued*

induced; blue: repressed; white: no change; grey: ortholog absent in species. (**B**)–(**F**) Correlation in expression decreases with phylogenetic distance. Shown are scatter plots relating—for each pair of species—their estimated phylogenetic distance (y axis) and the correlation between their matching global expression profile (x axis) at a matching physiological time point (noted on top). The legend shows the clade to which the pair belongs (if the same) or 'other' (if from different clades). Branch length was scaled by the maximum branch length to range from 0 to 1. (**B**) Lag, (**C**) Late Log (LL), (**D**) Diauxic Shift (DS), (**E**) Post Shift (PS), (**F**) Plateau (PLAT). The line in each plot is the least squares fit. (**G**) Shown is the average Pearson's correlation between pairs of species of the global expression profiles for each physiological time point.

The following source data and figure supplements are available for figure 3:

**Source data 1**. Experimental design and correlation among biological replicates.

**Figure supplement 1**. Conservation of growth-rate regulated gene expression.

difference between species (MS, DAT and AR, unpublished results), and additional profiling in some of the most divergent species at finer increments did not change this finding (data not shown), ruling out this possibility. Nevertheless, some functional groups of genes do exhibit conserved induction or repression in the lag phase across most species (e.g., growth genes such as mRNA splicing genes are enriched among the highly induced genes [$p<10^{-5}$, FDR, 'Materials and methods'] possibly to support splicing of transcripts encoding ribosomal proteins).

## The transcriptional response consists of five major modules in all species

To automatically trace the evolutionary trajectory of gene regulation, we used a novel probabilistic algorithm, Arboretum (*Roy et al., 2013*) ('Materials and methods'), to infer modules of co-expressed genes in each extant species and to reconstruct the ancestral modules from which they were derived. Arboretum uses a generative probabilistic model of the evolution of module membership of a given gene, starting from the last common ancestor (LCA) of a set of species and probabilistically propagating this membership down the branches of the phylogenetic tree to the leaf nodes. Arboretum uses the phylogeny to link the module IDs at each phylogenetic point to the modules of the LCA, and uses the structures of gene trees (as defined from genome sequences; *Wapinski et al., 2007a*) during module identification. This allows Arboretum to simultaneously infer ancestral and extant module assignments, and to systematically handle complex orthology and paralogy relationships that arise from gene duplication and loss. Arboretum allows genes to change their module assignment during evolution, but at any ancestral or extant species, every gene (if present in that species) must be assigned to exactly one module for a particular response.

We first applied Arboretum only to the transcriptional profiles of those genes that had no duplication events, but could have been lost or be lineage-specific (*Table 1*; 'Materials and methods'), and found that in each species the data is best explained by five expression modules (*Figure 4*, Modules 1–5, *Figure 4—source data 1*), capturing 65–70% of the variation ('Materials and methods'): Module 1 (strong repression, 'growth' genes, e.g., for transcription and translation), Module 2 (mild repression, cell division genes), Module 3 (little or no change, cell morphogenesis genes), Module 4 (mild induction, proteasomal and mitochondrial genes), and Module 5 (strong induction, stress, respiration and carbohydrate metabolism genes, *Supplementary file 1*).

Modules conserve their gene members relative to those of their immediately ancestral internal vertex in the tree ('immediate ancestor'), inversely to phylogenetic distance. To quantify this, we defined an Ancestral Module Conservation Index (AMCI; *Figure 5A*; 'Materials and methods') based on the probability with which a gene in a species conserved its module assignment from its immediate ancestral module. Extant and ancestral species have relatively high AMCIs (mean 0.82 ± 0.137 standard deviation [SD]), indicating conserved module assignment (*Figure 5B*), and highlighting the consistency of our experimental design. AMCI values are inversely related to the length of the branch connecting a species and its immediate ancestor (*Figure 5C*, Pearson's correlation $r = -0.68$, $p<1.14 \times 10^{-4}$).

## The growth and stress modules are the most phylogenetically conserved

To assess the tendency of individual modules to diverge across the entire phylogeny, we traced the module membership of every gene through the phylogenetic tree from the LCA (A14) to the extant

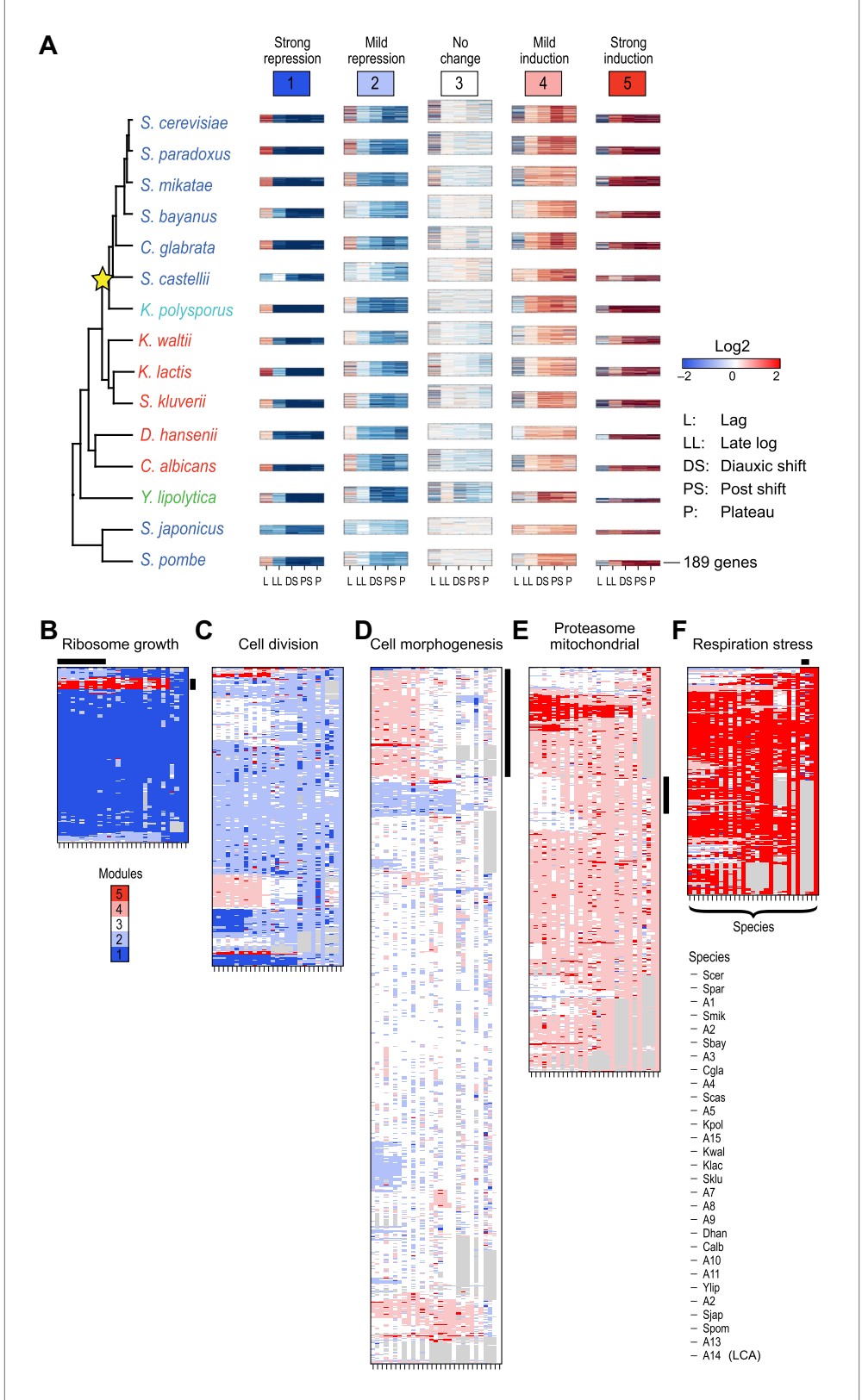

**Figure 4**. Arboretum reconstruction of expression module evolution (Analysis 1). (**A**) Five expression modules identified by Arboretum in the transcriptional response to glucose depletion. Each row corresponds to a species (tree, left) and each major column to a module (1–5, labels top). Module labels are color coded by the regulation of *Figure 4. Continued on next page*

*Figure 4. Continued*

the module's genes following depletion, as noted on top, from bright blue (Module 1) for strong repression to bright red (Module 5) for strong induction. Each module's height is proportional to the number of genes in the module. The five columns in each module are the expression levels at lag (L), late log (LL), diauxic shift (DS), post-shift (PS), and plateau (P) relative to mid-log phase. Red: induced; blue: repressed; white: no change. (**B**)–(**F**) Module assignments in all extant and ancestral species (see *Figure 5B* for ancestral node assignment). Each matrix corresponds to the genes in one of the five modules in the LCA (A14) (**B**: Module 1; **C**: Module 2; **D**: Module 3; **E**: Module 4; **F**: Module 5), and shows the module assignment of these genes in each of the extant and ancestral species from *S. cerevisiae* (leftmost column) to the LCA (rightmost column). The biological functions listed at the top of each module are representative labels chosen based on Gene Ontology terms enriched in all species in that module (**Supplementary file 1**). The range of FDR p values and fraction of genes in each module are as follows: Module 1: Ribosome biogenesis, p<5.28 × 10$^{-48}$ to 1.25 – 10$^{-119}$, fraction 37.3–61.6%. Module 2: cell division, p-value<3.51 × 10$^{-02}$ to 4.52 × 10$^{-02}$, fraction 9–33.6%. Module 3: cell morphogenesis, p<4.64 × 10$^{-02}$ to 4.95 × 10$^{-02}$, fraction 6.5–81%. Module 4: mitochondrial, p<3.20 × 10$^{-02}$ to 4.90 × 10$^{-02}$, fraction 2.4–37.9%; proteasome, p<3.85 × 10$^{-04}$ to 3.97 × 10$^{-02}$, fraction 1.6–15%. Module 5: respiration p<4.77 × 10$^{-02}$ to 4.8 × 10$^{-02}$, fraction 32.6–58.9%; response to stress, p<4.75 × 10$^{-02}$ to 4.86 × 10$^{-02}$, fraction 2.6–13.7%. Module assignment in each species is marked by a color code, as in the top of panel A (bright blue: Module 1; light blue: Module 2; white: Module 3; pink: Module 4; red: Module 5). Species are ordered by post-fix ordering (left-child, right-child and parent) of the species tree, as marked on the legend (bottom). Black bars indicate points of phylogenetically coherent divergence in expression of orthologous genes, as discussed in the text.

The following source data are available for figure 4:

**Source data 1**. Orthogroups included in the Aboretum run for analysis 1.

---

species (*Figure 4B–F*). Genes assigned in the LCA to Modules 1 (*Figure 4B*, blue entries) and 5 (*Figure 4F*, red entries) tend to persist in those modules (Module Stability 94% ± 0.5% and 84% ± 1.2%, respectively, 'Materials and methods'). Conversely, Modules 2, 3, and 4 (*Figure 4C–E*) are less stable, with more frequent reassignment of their member genes to other modules. Module 1 and 5 are the least dynamic as reflected by the average proportion of genes that join (11% ± 1.6% and 17% ± 1.7%, respectively) or leave each module (6% ± 0.4% and 16% ± 1%, respectively) between each pair of ancestor and child species in the phylogeny (Module Expansion Index and Module Contraction Index, respectively, 'Materials and methods'; *Figure 5D,E*).

## Reassignment of coherent functions between modules corresponds to changes in lifestyle and ploidy

Many coordinated phylogenetic transitions of functionally-related genes are consistent with differences in lifestyle between species. For example, cell cycle and DNA replication genes are reassigned in a coordinated fashion from Module 3 in the LCA to Modules 4 and 5 in *Schizosaccharomyces* (*Figure 6A*), consistent with the unique nutrient control of mating in this clade (*Nadin-Davis and Nasim, 1990*). In another case, both ploidy- and meiosis-related genes (*Figure 6B*) are reassigned from the strongly induced Module 5 in the typically haploid pre-WGD species to the unchanged Module 3 or repressed Module 2 in the typically diploid post-WGD species. This is likely due to shifts in nutrient control of mating and meiosis post-WGD (*Barsoum et al., 2011*). Arboretum infers that the LCA (A14) followed the haploid, Module 5, pattern (*Figure 6B*, middle panel, A14 column).

## Concerted reassignment of mitochondrial, respiratory and amino acid and sulfur metabolism genes associated with the evolution of respiro-fermentation

Particularly substantial transitions in module assignment are associated with changes in carbon lifestyle during the transitions to respiro-fermentation post-WGD and in *Schizosaccharomyces*. These include genes with diverse mitochondrial functions, including mitochondrial ribosomal protein genes (mRPs) that are reassigned from ancestral Modules 1 or 2 to Modules 3 or 4, independently at the WGD and in the *Schizosaccharomces* ancestor (A13) (*Figure 6C*). This is consistent with previous hypotheses that respiro-fermentation is the derived state (*Piskur et al., 2006*). Similar changes have occurred in the expression of oxidative phosphorylation genes (*Figure 6D*), albeit with a more gradual shift from

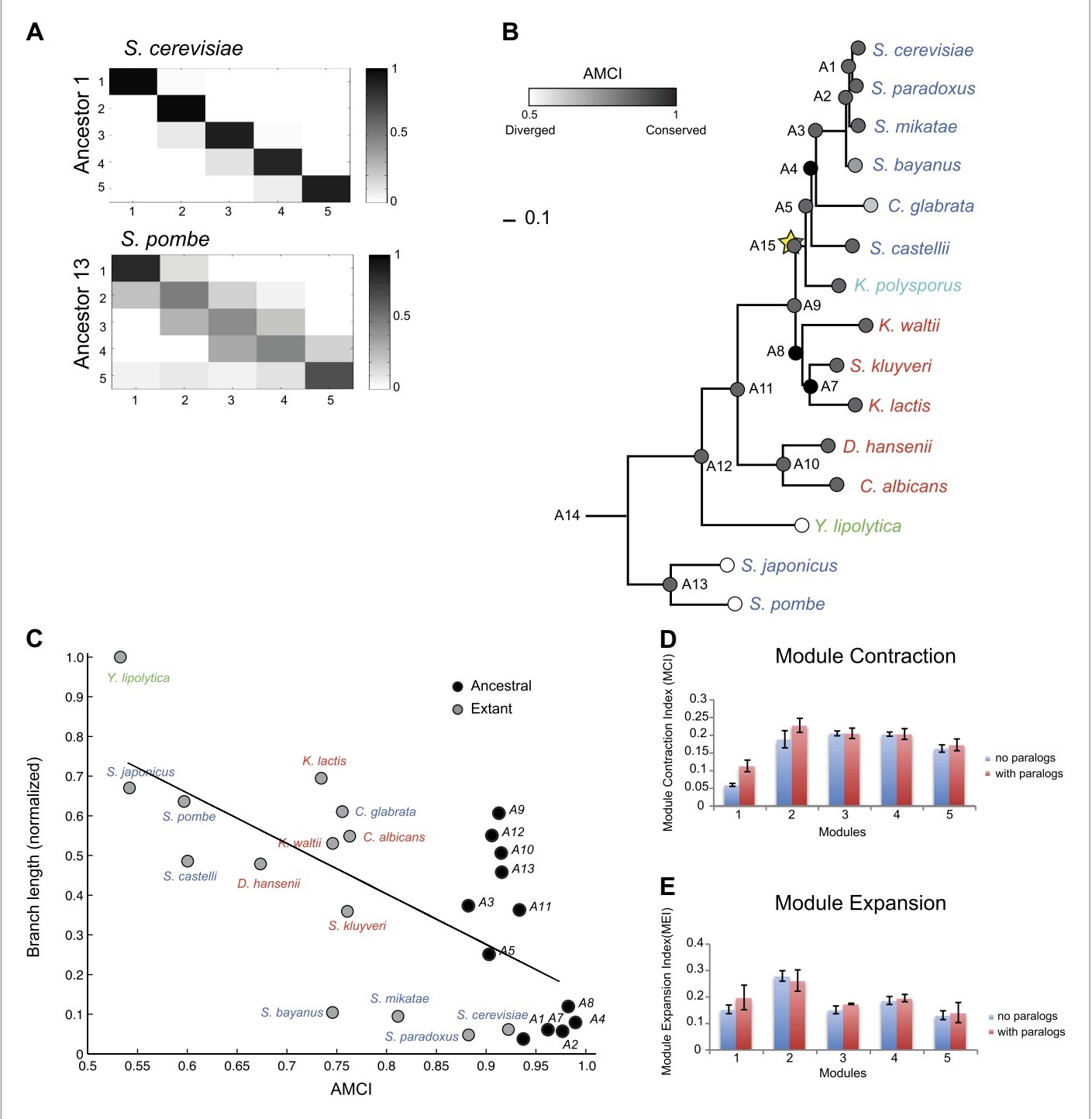

**Figure 5**. Conservation of modular organization. (**A**) Module transition matrices. Shown are examples of transition matrices estimated by Arboretum for two species (*S. cerevisiae*, top and *S. pombe*, bottom). Each matrix specifies, for each module in each child species (columns), the probability with which a gene conserved its module assignment in that species' immediate ancestor (rows), or was reassigned to another module. Columns: modules of the child species, rows: modules of the ancestor species. Probabilities are color coded from black (1) to white (0). Strong diagonal elements indicate high conservation with the immediate ancestor. The AMCI is calculated as the mean of the diagonal entries. (**B**) The Ancestral Module Conservation Index (AMCI). Shown is the AMCI, ranging from 0: least conserved (white circles) to 1: most conserved (black circles), for each extant and ancestral species. Tree is drawn to scale and species are color coded by carbon lifestyle as in *Figure 1A*. (**C**) AMCI decreases with increased phylogenetic distance. Shown is a scatter plot of the relationship, for each extant (grey) and ancestral (black) species, between its phylogenetic distance to its immediate

*Figure 5. Continued on next page*

*Figure 5. Continued*

ancestor (branch length, y axis) and its AMCI (x axis). Branch length is scaled by the maximum value to range between 0 and 1. The correlation between branch length and AMCI is −0.68 (p≤1.13 −× 10⁻⁴). The regression line is plotted. (**D**) and (**E**) Expansion and contraction of modules. Shown are the mean Module Contraction Index (MCI, **D**) and mean Module Expansion Index (MEI, **E**) for each Arboretum module (x axis), based on the proportion of genes that respectively leave or join each module at each phylogenetic point. Blue and red indicate the modules from Arboretum runs with only no duplicates (no paralogs) and including duplicates (with paralogs), respectively. Error bars were estimated from five Arboretum runs with different initializations.

repression in *Y. lipolytica* to late induction in *Kluyveromyces* to strong induction post-WGD (***Figure 6D***, left panel). This is associated with concomitant changes in nucleosome organization and anti nucleo-somal polydA:dT sequences in the promoters of oxidative phosphorylation and mRP genes (***Field et al., 2008***; ***Tsankov et al., 2010***), which occurred convergently post-WGD and in *Schizosaccharomyces* (***Figure 7A,B***).

In addition, the promoters of mRP and respiration genes have lost the binding site for the transcription factor Sfp1 (previously identified as the 'Rapid Growth Element', RGE; ***Ihmels et al., 2005***) in many respirofermentative species (***Figure 6—figure supplement 1A***). The Sfp1 binding site is strongly conserved in 'growth' (Module 1) genes across all species (except *S. japonicus*) and serves as an ancient regulatory tether of the growth module (p<0.05, Hyper-geometric test; ***Figure 6—figure supplement 1B***). This site was lost from mRP gene promoters in post-WGD species (***Ihmels et al., 2005***; ***Tanay et al., 2005***) and *S. japonicus* (***Figure 6—figure supplement 1A***). In *S. pombe* and *K. polysporus* some genes have lost the site whereas others retain it, where the number retained is an intermediate between that in post-WGD species and in respiratory species (***Jiang et al., 2008***). The Sfp1 motif is also enriched in oxidative phosphorylation gene promoters in some of the respiratory species (Hyper-geometric, p<0.05, ***Figure 6—figure supplement 1A***).

Indeed, more surprisingly, purine metabolism and amino acid catabolism genes in pathways that feed nucleotide synthesis (***Figure 8A***) are reassigned from mildly repressed or unchanged Modules 2 and 3 pre-WGD to the strongly induced Module 5 post-WGD (***Figure 6E***). These regulatory changes were not previously recognized, to the best of our knowledge, and are reminiscent of those observed in the Warburg effect (***Vander Heiden et al., 2009***), a cancer metabolic state analogous to respiro-fermentation (see 'Discussion'), including the induction of the glycine biosynthetic pathway that was recently shown to be correlated with proliferation rates across different cancers (***Jain et al., 2012***). Unlike mRP and oxidative phosphorylation genes, this re-assignment is unique to the post-WGD species and does not occur in *Schizosacchromyces* (***Figure 6E***). It is associated with the shifting of binding sites for key activators of amino acid and purine biosynthesis genes (e.g., Bas1, Met32, Gzf3, Gln3, Met28, Met4, and Gat1) from nucleosome-free to nucleosome-occluded positions (***Tsankov et al., 2010***) in rich media post-WGD (***Figure 7C***). Conversely, the phylogenetic pattern of Sfp1 motif enrichment is less conclusive in these genes (***Figure 6—figure supplement 1A***).

## Paralogous genes significantly enhance module divergence, especially at the whole genome duplication and the *Schizosaccharomyces* ancestor

Changes in expression following gene duplication have previously been proposed to contribute to evolutionary innovation (***Gu et al., 2004***; ***Ihmels et al., 2005***; ***Guan et al., 2007***; ***Tirosh and Barkai, 2007***; ***Wapinski et al., 2007b***). In particular, several studies (***Teichmann and Babu, 2004***; ***Gu et al., 2004***, ***2005***; ***Conant and Wolfe, 2006***) showed increased transcriptional divergence of paralogous genes in *S. cerevisiae* compared to non-duplicated genes, consistent with a model of accelerated transcriptional evolution following duplication (***Gu et al., 2005***). However, since such studies typically compared the expression of paralogous genes in only one or two species (***Teichmann and Babu, 2004***; ***Gu et al., 2004***, ***2005***; ***Conant and Wolfe, 2006***; ***Guan et al., 2007***; ***Tirosh and Barkai, 2007***), they could not reliably infer the ancestral state, and had to rely on different assumptions, for example that similarity between paralogs in a single species reflects their ancestral state (***Gu et al., 2005***; ***Teichmann and Babu, 2004***) or that expression divergence should be interpreted within the context of sequence evolution (***Gu, 2004***).

To systematically test the potential impact of gene duplication on the transcriptional response to glucose depletion, we applied Arboretum to an expanded set of 3676 orthogroups that incurred one duplication event ('Materials and methods'; ***Table 1***). The modules identified after inclusion of paralogs

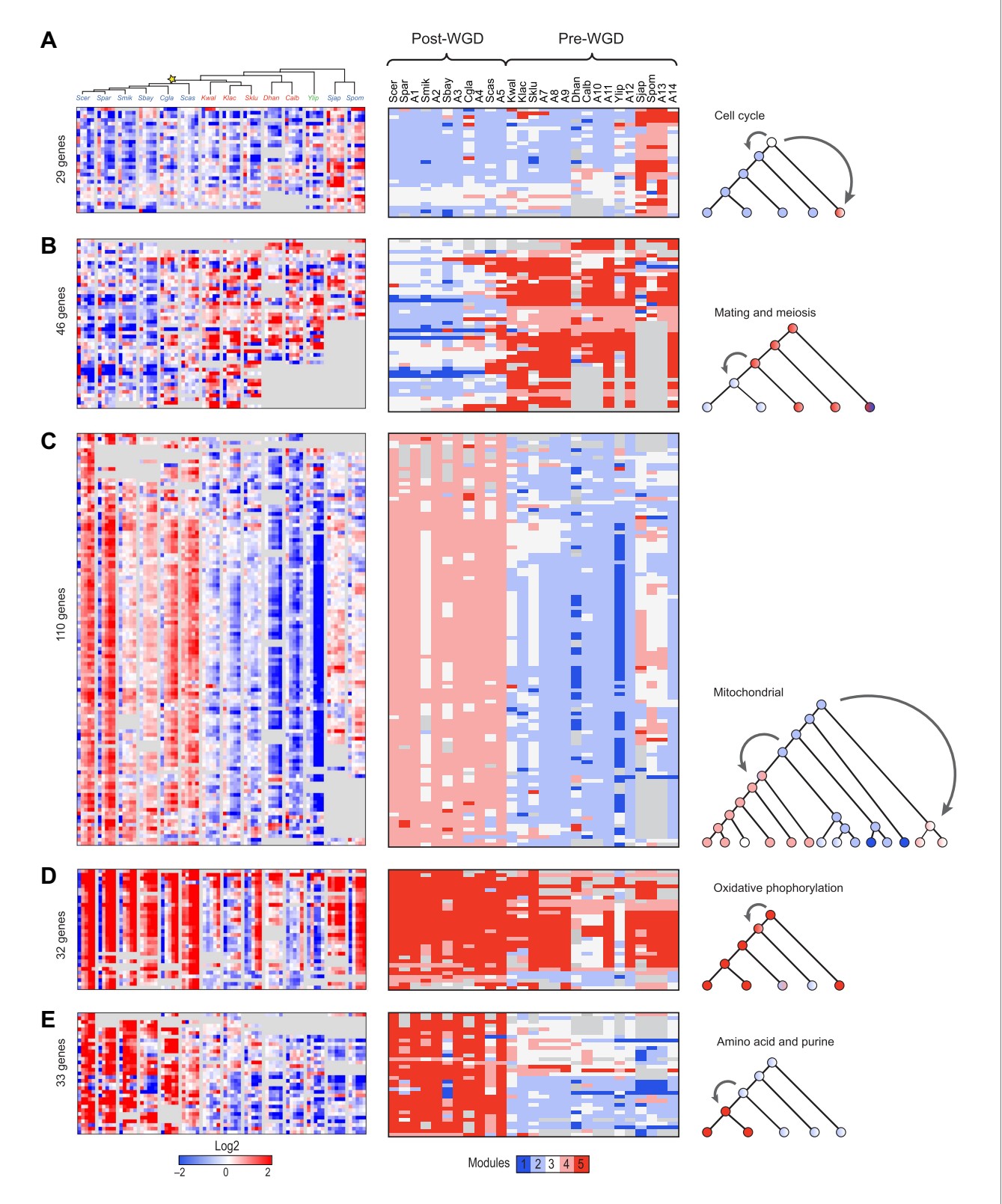

**Figure 6**. Conservation and rewiring of coherent functions across modules. Shown are expression (left), Arboretum module assignments (middle) and a cartoon of the phylogenetic transition (right) for gene sets with coherent phylogenetic patterns. Each expression matrix is formatted as in *Figure 3A*, and each module assignment matrix as in *Figure 4B–F*. (**A**) Cell cycle genes, (**B**) mating and meiosis related genes, (**C**) mitochondrial genes, *Figure 6. Continued on next page*

*Figure 6. Continued*

(**D**) oxidative phosphorylation genes, (**E**) amino acid and purine metabolism genes. Each module shows all the genes with a given phylogenetic pattern, and their labels (e.g., mitochondrial) were manually generated based on enrichment of GO terms.
The following figure supplements are available for figure 6:

**Figure supplement 1**. Enrichment of Sfp1 binding sites.

exhibited the same major phenotypic patterns of expression (*Figure 9*, *Figure 9—source data 1*, *Supplementary file 2*), but had a lower AMCI (paired *t*-test, p<0.03, *Figure 10A*).

This increased divergence is specifically associated with paralogous genes. Genes that have a paralog were more likely to change their module assignment than those present in a single copy (KS test, p<1.5 × 10$^{-107}$). Furthermore, controlling for any differences in function between duplicated and non-duplicated genes, paralogous genes are much more likely to be reassigned after duplication than their corresponding pre-duplication orthologs (*Figure 10D,E*, KS test, p<1 × 10$^{-18}$).

This increased divergence was most prominent at the WGD ancestor (A5) and the *Schizosaccharomyces* ancestor (A13) (*Figure 10A*). Both ancestors also have a larger number of retained paralog pairs compared to other ancestral species and reside at the points of the (independent) evolution of respiro-fermentation. Consistently, the reassigned paralogs are enriched in carbon metabolism genes (*Figure 10—source data 1*).

## Enhanced regulatory divergence during a short window post-duplication

Most paralogs are reassigned within a short window following their duplication. In most cases, divergence in paralog assignment is already apparent at the immediate phylogenetic point of the duplication event (at the phylogenetic resolution of this study). For example, ~60% of the paralog pairs that arose at the ancestor of the *Candida* clade (A10) and that will eventually change their assignment, do so at that ancestor. Similar trends are observed for paralogs that arose at most other ancestors (e.g., A11, 54%; A9, 48%). Overall, of all the paralog pairs that arose sporadically (not in the WGD) in the phylogeny, and where at least one member of the pair was subsequently reassigned to a different module, 53% do so 'at' the point at which they arose. Furthermore, such sporadically duplicated genes are significantly enriched (Hyper-geometric test, p<0.05,) among all the reassigned genes (duplicated or not) at their point of duplication, but not at later phylogenetic points, when they behave comparably to non-duplicated genes (*Figure 10B*). This suggests a short 'window of opportunity' in regulatory innovation, specifically facilitated by paralogous genes shortly after they are duplicated but not later, and provides strong experimental support to previous theoretical models (*Gu et al., 2005*).

## A prolonged effect of WGD paralogs on regulatory divergence

Paralogs that arose in the WGD (*Figure 10B*, A5) are a notable exception to some of these trends, and may affect divergence for a longer period. While a similar proportion (54%) of those that diverge do so as early as they arise, they continue to contribute significantly to the divergence in module assignment at later points compared to non-duplicated genes (*Figure 10B*, orange circles, Hyper-geometric test, p<0.05). This distinction is not simply due to the finer phylogenetic resolution in the post-WGD clade in our study: the same trend is observed with eight of the 15 species with comparable resolution in the pre- and post-WGD clades (*Figure 11A,B*). This extended window may be explained by the theoretical argument (*Lynch and Katju, 2004*) that sporadically duplicated genes—but not WGD duplicated genes—may not be duplicated with the full set of ancestral regulatory elements and may move to novel genomic locations (*Lynch and Katju, 2004*), thus resulting in immediate divergence from the ancestral state.

## Most duplication events lead to assignment of the paralogs to two different modules

Paralogs may diverge either by generation of new functions (neo-functionalization) or by partitioning of several ancestral functions (sub-functionalization) between the paralogs through complementary

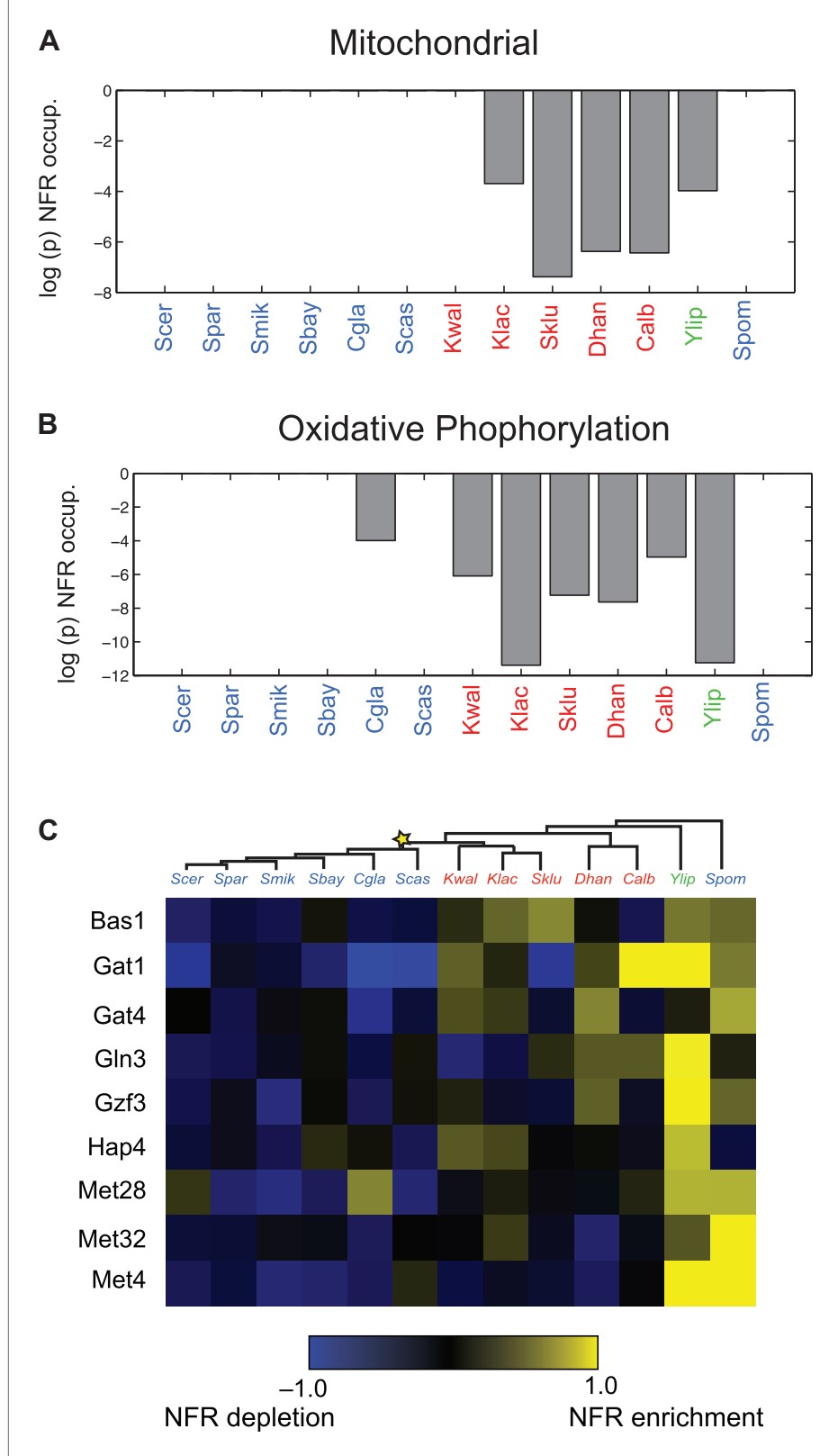

**Figure 7**. Changes in chromatin organization in mitochondrial, oxidative phosphorylation and amino acid metabolism genes. Shift in NFR occupancy in re-wired respiratory genes (**A** and **B**). Shown are the logarithm of the p value of the KS-test (y axis) used to test if the genes in a given set (mitochondrial genes, **A**, and oxidative phosphorylation
*Figure 7. Continued on next page*

*Figure 7. Continued*

genes, **B**) have a significantly lower nucleosome occupancy at their 5'NFRs than that of all genome genes in each of 13 species (x axis) with nucleosome positioning data from *Tsankov et al. (2010)* and *Xu et al. (2012)*. (**C**) Evolutionary repositioning of binding sites for key amino acid TFs relative to NFRs. For each of 13 species (columns, tree), shown are the enrichment (yellow) or depletion (blue) in NFRs of binding sites for several amino acid and purine metabolism TFs (rows) whose sites are depleted from NFRs in post-WGD species and enriched in pre-WGD species. The intensity of the color is proportional to the z-score estimated for each regulator from the fraction of all its binding sites that are in the NFR. Each row is centered by its mean value (see 'Materials and methods').

degeneration (the Duplicate-Degeneration-Complementation [DDC] model; *Lynch and Force, 2000*). In the context of gene regulation, these events can in principle manifest either as changes in the expression of 'targets' (e.g., through *cis*-regulatory changes) or by changes in regulator function (e.g., in a transcription factor's responsiveness to signals or its binding site specificity). Previous studies of *S. cerevisiae* paralogs (*Teichmann and Babu, 2004*; *Conant and Wolfe, 2006*; *Tirosh and Barkai, 2007*) showed indirect evidence for both sub- and neo-functionalization, but had to make strong assumptions on the ancestral state with little experimental data.

To assess the extent of regulatory neo-functionalization of targets, we compared the module assignment of the two paralogs at the point of duplication to that of their immediate pre-duplication ancestor (four possible fates, *Figure 10C*, 1–4). We focused only on this earliest transition since the majority of paralogs (53%) diverge first there, and since the pre-duplication module assignment provides a well-defined, identical reference point for both duplicates, whereas comparisons at later point do not afford a common reference.

Our phylogenetic analysis allowed us to refine divergence events beyond the previously defined categories. Most paralogs (66% of all pairs) have at least one member diverging from the ancestral module. This includes paralogs with distinct fates, either due to 'classical' neo-functionalization (27% of all pairs), where one paralog maintains the ancestral assignment and the other is assigned to a different module (e.g., the purine biosynthesis genes *URA8* and *URA7*, *Figure 10C*, case 2), or due to asymmetric divergence (12%), where each paralog is reassigned to a distinct module, each different than the ancestral one (e.g., the disulphide isomerases *EUG1* and *PDI1*, *Figure 10C*, case 3). In the remaining cases (27%) there is symmetric divergence, where both paralogs are reassigned to the same module, distinct from the ancestral one (e.g., the ribonucleotide-diphosphate reductases *SER3* and *SER33*, *Figure 10C*, case 4). Only in a minority of paralog pairs (34%) the ancestral assignment of both copies is conserved (e.g., the ribosome biogenesis genes *UTP5* and *UTP9*, *Figure 10C*, case 1). These classes could be easily confounded in previous studies relying on paralogs from a single species (e.g., 'conserved' and 'symmetrically diverged' would be indistinguishable). The pre-duplication orthologs of genes from the neo-functionalized class are most prominent in the 'unchanged' Module 3 and mildly induced Module 4 (Hyper-geometric test, $p < 10^{-3}$), suggesting that genes that were not regulated in glucose depletion prior to duplication, 'gain' such regulation de novo post-duplication, for at least one paralog. Neo-functionalization has also contributed to the regulatory divergence of key genes in carbon metabolism during the evolution of respiro-fermentation (*Figure 8B*), consistent with previous studies (*Conant and Wolfe, 2007*).

## The same evolutionary patterns are also apparent in the heat shock response

To assess the generality of our findings, we compared the key patterns discussed above to corresponding ones observed in the heat stress response in eight of the same species (*Roy et al., 2013*). As in the response to glucose depletion, similarity in heat stress profiles is inversely proportional to phylogenetic distance (*Figure 11C–F*), and the strongly repressed Module 1, which consists of 'growth' genes, is the most conserved (*Roy et al., 2013*). Furthermore, the evolutionary trajectories of paralogous genes—including the higher reassignment of paralogs, the distinct reassignment capacity of sporadic and large-scale duplications, and the preponderance of reassignment of only one of two paralogs—are highly similar between heat shock and glucose depletion (*Figure 11A,B*). This suggests that these are general evolutionary principles, at least in this phylogeny.

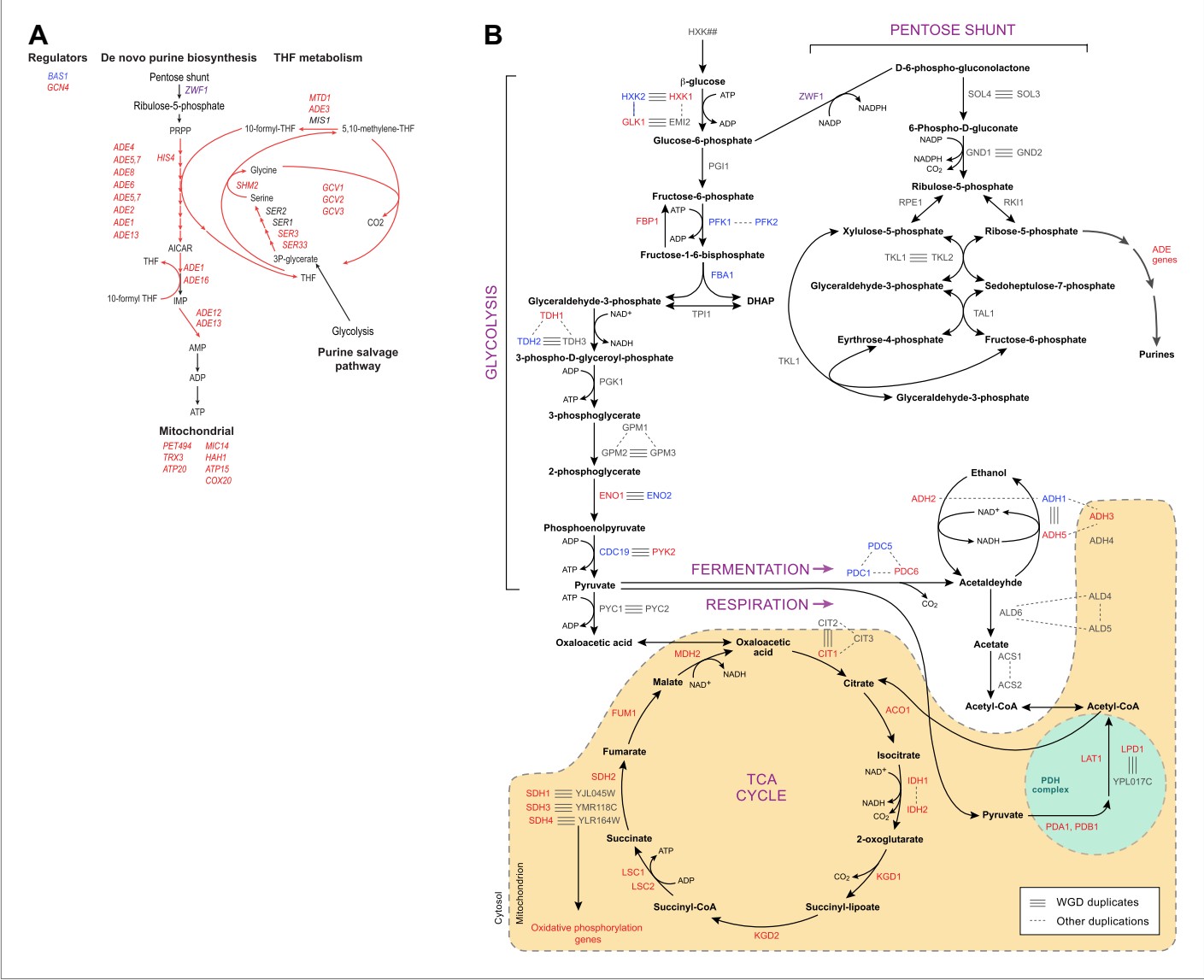

**Figure 8**. Purine and amino acid metabolic pathways are linked to carbon metabolism. (**A**) Shown are the set of metabolic reactions in *S. cerevisise* associating purine biosynthesis and salvage and amino acid metabolism with carbon metabolism, and two key transcriptional regulators (left). Mitochondrial genes link respiration to purine metabolism. Glycolysis is linked to purine salvage by the metabolic intermediate 3-P-glycerate. De novo purine metabolism is linked to the pentose shunt through ribulose-5-phosphate. The genes in red are induced post-shift in *S. cerevisiae* and other post-WGD species, but their orthologs are repressed in pre-WGD species. Both *Schizosaccharomyces* species have three copies of *ZWF1* (purple) that are strongly induced. (**B**) Shown are the major carbon pathways involved in the fermentation or respiration of glucose and their interconnectivity. Both WGD and other duplicate genes in each pathway are indicated. The genes in red are induced post-shift in *S. cerevisiae* and most of the other post-WGD species while those in green are repressed similar to their pre-duplication orthologs. Differences in *trans* regulators may further contribute to the reassignment of their targets between modules. While many of the regulators of glucose repression in *S. cerevisiae* are present across the phylogeny (*Flores et al., 2000*), the regulation of some has changed at the WGD and at the ancestor of the *Schizosaccharomyces*, consistent with the reassign-ment of their targets. For example, the glucose repressing *MIG* genes and the *TUP1-CYC8* complex are strongly repressed following glucose depletion in most post-WGD species, whereas some respiration activators are strongly induced (*CAT8* and *HAP2,4,5* and *SIP2* post-WGD, *HAP2*, *MOT3*, and *SIP2* in *S. pombe*, data not shown). We observed no such changes in the expression of known regulators of amino acid and purine metabolism (data not shown). In some cases, duplication of key regulators followed by reassignment to a new module may have further contributed to new regulatory functions. For example, *TPK1* and *TPK3* are two WGD-derived paralogs encoding catalytic subunits of PKA, a major regulator of carbohydrate metabolism and stress responses (*Zaman et al., 2008*). *TPK1* in strongly induced in the *sensu stricto* species, as is the single *TPK* gene in the *Schizosaccharomyces. TPK3* is repressed in those species, conserving the expression pattern of its ortholog in all the respiratory pre-duplication species (data not shown).

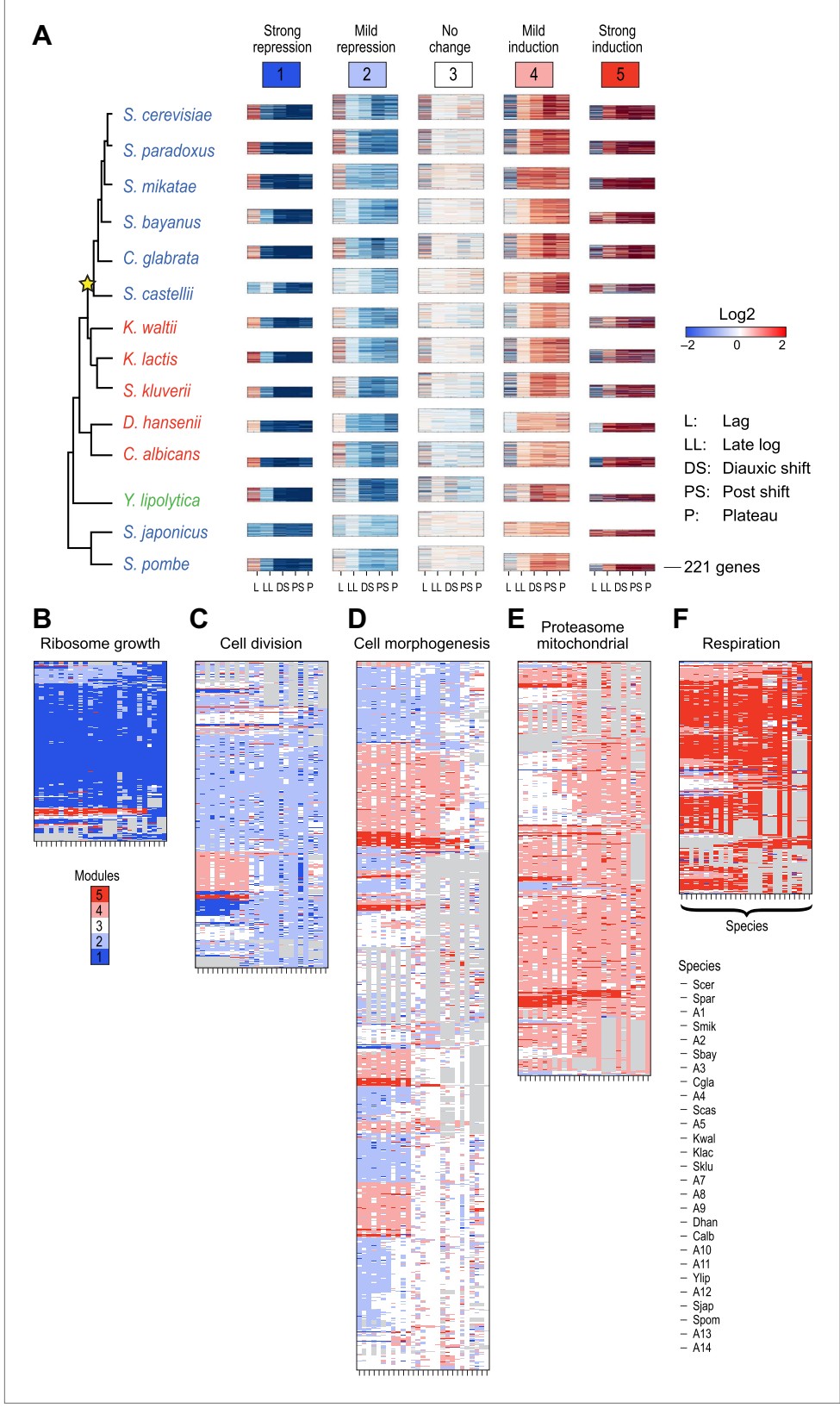

**Figure 9**. Arboretum reconstruction of expression module evolution in the presence of paralogous genes (Analysis 2).
(**A**) Five expression modules identified by Arboretum in the transcriptional response to glucose depletion, when
paralogous genes are included in the run. Each row corresponds to a species (tree, left) and each major column to
*Figure 9. Continued on next page*

*Figure 9. Continued*

a module (1–5, labels top). Modules labels are color coded by the regulation of the module's genes following depletion, as noted on top, from bright blue (Module 1) for strong repression to bright red (Module 5) for strong induction. Each module's height is proportional to the number of genes in that module. The five columns in each module are the expression levels at lag (L), late log (LL), diauxic shift (DS), post-shift (PS), and plateau (P) relative to mid-log phase. Red: induced; blue: repressed; white: no change. (**B**)–(**F**) Module assignments of all extant and ancestral species. Each matrix corresponds to the genes in one of the five modules in the LCA (**B**: Module 1; **C**: Module 2; **D**: Module 3; **E**: Module 4; **F**: Module 5), and shows the module assignment of these genes in each of the extant and ancestral species from *S. cerevisiae* (leftmost column) to the LCA (rightmost column). The biological functions listed at the top of each module are general classifiers based on Gene ontology terms enriched in all species in that module (**Supplementary file 2**). The range of FDR p values and fraction of genes in each module are as follows: Module1: ribosome biogenesis, $p<1.07 - 10^{-52}$ to $1.56 \times 10^{-112}$, fraction 32–53%. Module2: cell division, $p<3.13 \times 10^{-02}$ to $4.69 \times 10^{-02}$, fraction 10.2–32%. Module 3: cell morphogenesis, $p<4.48 \times 10^{-02}$ to $4.56 \times 10^{-02}$, fraction 22–78.7%. Module 4: mitochondrial, $p<2.47 \times 10^{-02}$ to $3.36 \times 10^{-02}$, fraction 2.3–36.2%; proteasome, $p<2.7 \times 10^{-03}$ to $5.48 \times 10^{-03}$, fraction 1.3–13.1%. Module 5: respiration, $p<4.2 \times 10^{-02}$ to $4.43 \times 10^{-02}$, fraction 34.9–55%. Module assignment in each species is marked by a color code, as in the top of panel a (bright blue: Module 1, light blue: Module 2, white: Module 3, pink: Module 4, red: Module 5). Species are ordered by post-fix ordering (left-child, right-child and parent) of the species tree, as marked on the legend (bottom).
The following source data are available for figure 9:

**Source data 1**. Orthogroups included in the Aboretum run for analysis 2.

The additional condition allows us to further refine the classification of paralog fates (*Figure 12*). For example, some paralogs whose expression is 'conserved' in glucose depletion, may be reclassified as neo-functionalized when heat shock data is also considered, for example, due to a new regulation of one paralog in heat shock. Indeed, considering both responses, the conserved class was reduced from 34% to 28%, and the majority (70%) of paralogs were in classes where either one (38%, *Figure 12B,E*) or both (32%, *Figure 12C,E*) were neo-functionalized in at least one response. Surprisingly, only 2% of all paralog pairs were in the sub-functionalized class (*Figure 12D,E*).

## Discussion

Here, we used comparative functional genomics of transcriptional profiles measured during growth of 15 species of *Ascomycota* on glucose to identify several principles in the evolution of gene expression in a complex phylogeny. Using yeasts allowed us to control growth conditions and closely monitor comparable physiological responses. The substantial overall conservation we observed in the response—exceeding that reported in other studies (*Tirosh et al. 2011*), indicates the robustness of our experimental strategy. However, despite this conservation, we found that the degree of correlation is inversely related with the phylogenetic distance between the species.

Several lines of evidence suggest that these global differences in expression profiles are not simply dominated by the effect of growth rates or of 'carbon metabolism' lifestyle. First, the expression profiles in each species for a set of genes recently identified as regulated by growth rate in *S. cerevisiae* (*Brauer et al., 2008*) are similar across species in the phylogeny (*Figure 3—figure supplement 1A*), consistent with our physiological sampling. Indeed, removal of these genes did not significantly change the correlation of global expression profiles (p>0.1, paired *t*-test) at any of the points other than lag phase (*Figure 3—figure supplement 1B–F*, paired *t*-test, $p<2.02 \times 10^{-8}$). Notably, physiologically-matching profiles are significantly correlated even between distant species (e.g., Pearson's $r = 0.54$ between *S. cerevisiae* and *K. lactis* at late log), supporting our experimental design. Second, the global profiles of post-WGD and *Schizosaccharomyces* species that share a respiro-fermentative lifestyle are nonetheless most distant from each other (e.g., Pearson's $r = 0.43$ between *S. paradoxus* and *S. pombe* at late log).

To systematically compare the responses of the different species to glucose depletion we developed Arboretum, a probabilistic algorithm, which incorporates the species and gene phylogenetic relationships to identify modules in each species and to reconstruct the evolutionary trajectory of each gene's module assignment from the LCA (*Roy et al., 2013*). During the learning process, Arboretum

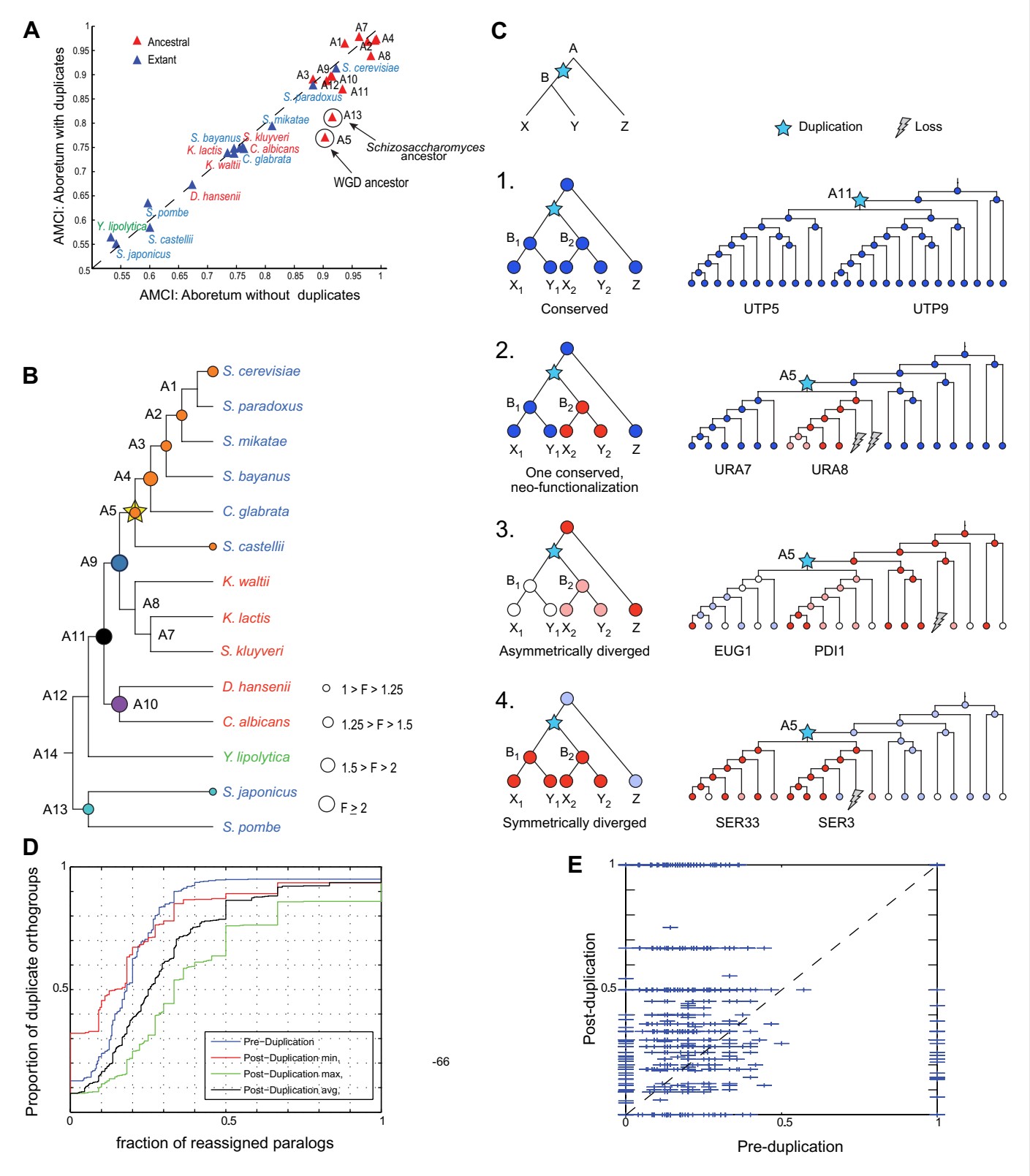

**Figure 10**. Regulatory evolution of paralogous genes. (**A**) Paralogous genes contribute to regulatory divergence. Shown in a scatter plot of the AMCI values for each extant (blue) and ancestral (red) species as estimated by Arboretum in a run without paralogs (Analysis 1) (y axis) vs a run with paralogs (x axis). Inclusion of paralogous genes lowers the AMCI, especially at the WGD and *Schizosaccharomyces* ancestors (arrows). (**B**) Enrichment of paralogous
*Figure 10. Continued on next page*

*Figure 10. Continued*

genes among reassigned genes. Shown is for each species (ancestral and extant) the fold enrichment (**F**) of paralogs (circle size) among genes reassigned at that species. Only points at which there are significantly more paralogs that switch than expected by chance are shown (Hyper-geometric p<0.05). Circles are colored by the phylogenetic point of gene duplication (cyan: A13, black: A11, purple: A10, blue: A9, white: WGD ancestor A5). (**C**) Four possible regulatory fates of paralogous genes following duplication, relative to their immediate pre-duplication ancestor. Left: cartoon gene trees (left) and illustrative examples from our analysis (right) representing the module assignment (circles) of each paralog and their pre-duplication ortholog in each extant and ancestral species. Module assignment is color coded as in *Figure 3* (Bright blue, light blue, white, pink, red from Module 1 to 5, respectively). Star: gene duplication. Lightning rod: gene loss. (1) Conserved: both paralogs (*UTP5* and *UTP9*) conserve the ancestral assignment (Module 1); (2) Neo-functionalization: one paralog (*URA7*) maintains the ancestral assignment (Module 1) and the other (*URA8*) is assigned to a different module (Module 5); (3) Asymmetric divergence: both paralogs (*EUG1*, *PDI1*) are reassigned to distinct modules (Module 3, Module 4) than the ancestral one (Module 5). (4) Symmetric divergence: both paralogs (*SER3*, *SER33*) are reassigned to the same module (Module 5), distinct from the ancestral one (Module 1). (**C**) Cumulative distribution of module reassignment of genes before and after their duplication. Because after duplication there are two paralogs, each with its own re-assignment value, we compare the minimum (red, $p<1 \times 10^{-4}$), maximum (green, $p<1 \times 10^{-66}$), and average (black, $p<1 \times 10^{-18}$) of the number of re-assignments after duplication, with the re-assignments before duplication (blue). (**D**) Scatter plots showing for each gene its degree of module reassignment before duplication (x axis) vs the average degree of module reassignment of the two paralogs after duplication (y axis). All module reassignments for a gene are normalized by the number the species in which the gene is present ('Materials and methods').

The following source data are available for figure 10:

**Source data 1**. Gene Ontology enrichment in sets of duplicate genes that diverged from the pre-duplication ancestor.

uses a soft probabilistic assignment of module membership allowing a gene to contribute to the mean expression profile of different modules. At the end of the learning, the gene is assigned to the single module with the highest probability. In the majority of cases, this 'hard assignment' is well supported in our dataset, such that the probability that a gene belongs to its assigned module is much higher than its probability to belong to the next-best module (*Figure 13A,B*) In some cases, however, a 'soft' probabilistic assignment may be preferable, to reflect the pleiotropic action of genes. A user can readily do this using the probabilities calculated by Arboretum.

Another key parameter that could impact our results is the number of modules. We chose a relatively small number (k = 5) as a conservative choice, which may under-estimate divergence relative to conservation, since genes with minor differences in expression between species would still likely belong to the same module. This choice increases our confidence in those divergence events that are highlighted, but may miss other divergence events. Several lines of evidence support the validity of our choice including the correspondence between the overall trends based on the direct expression levels (e.g., *Figure 3B–F*) and those based on inferred modules (e.g., *Figure 4C*), the substantial proportion of variance (65–70%) explained by 5 modules (*Figure 14*), and the lack of additional clear temporal patterns or distinct functional enrichments at higher *k*'s (*Figure 14— figure supplement 1*). Our companion manuscript (*Roy et al., 2013*) provides user guidelines for choosing *k*. Notably, Arboretum does not model dependencies between different time points, modeled each time point with a separate mean and variance. It is possible that by explicitly modeling temporal dependencies we can capture finer-grained dynamic differences. This is a direction of future research.

Arboretum traces module ancestry and handles duplication events naturally using the tree structure as part of the probabilistic generative process of module assignments. In particular, the module assignments at the point of duplication are two independent draws of module membership, one each for copy, ensuring that the orthogroup has the same module assignment before the duplication and allowing it the freedom to change assignment after duplication. We used these features to characterize the role of gene duplication in the evolution of gene regulation, substantially expanding on previous analyses based on comparing paralog expression in on one or two species. We found that paralogs significantly contribute to divergence, mostly during a short window of opportunity after duplication, with a more prolonged contribution from WGD paralogs. This may be explained by the fact that segmental duplication rarely preserves the entire regulatory region resulting in immediate expression divergence (*Lynch and Katju, 2004*) whereas duplicates from WGD events preserve the regulatory inputs and thus may take longer to diverge.

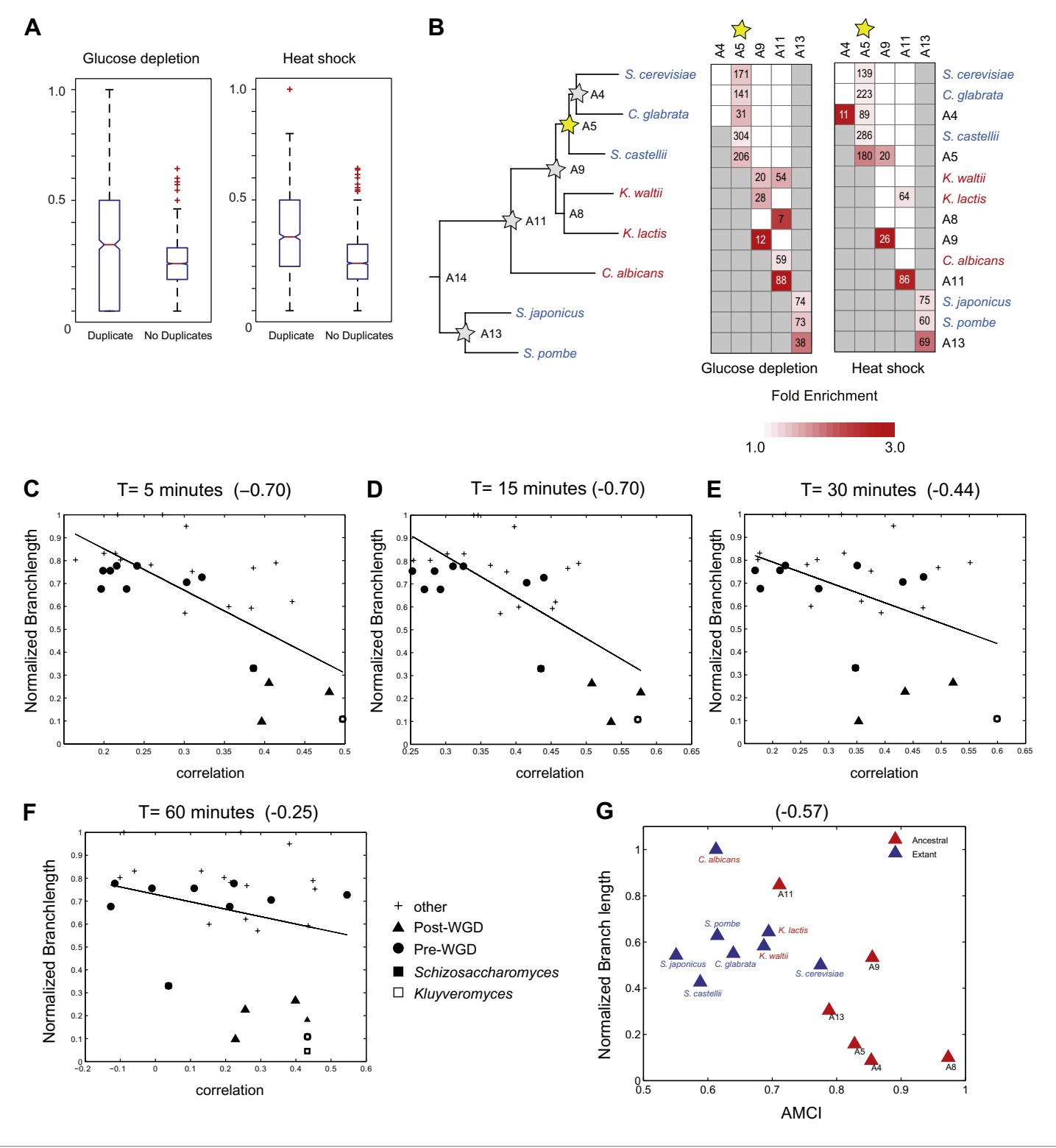

**Figure 11**. Similar evolutionary patterns in glucose depletion and heat shock. (**A**) Increased re-assignment of paralogous genes. Box-plots showing the fraction of module re-assignments for genes from orthogroups with duplication events (Duplicate, left) and without duplication events (Singleton, right). Red plus: outliers that are ±2.7 SD from the mean. (**B**) Enriched re-assignment of paralogous genes at different phylogenetic points. Shown are the fold enrichment of paralogous genes among all the reassigned genes (red, scale bar) at different phylogenetic points (rows) for *Figure 11. Continued on next page*

*Figure 11. Continued*
duplicates that arose at different ancestors (columns) for heat shock (left) and glucose depletion (right). The number in each cell represents the number of paralogous genes that arose at a given phylogenetic point (column) and were reassigned at a phylogenetic point (row). Numbers and fold enrichment are marked only at points with significantly more paralogs that are reassigned than expected by chance (Hypergeometric p<0.05). (C)–(F) correlation in expression decreases with phylogenetic distance. Shown are scatter plots relating—for each pair of species—their estimated phylogenetic distance (y axis) and the mean correlation between their matching global expression profiles (x axis) at matching time points (labeled on top). Legend shows the clade to which the pair belongs (if the same) or 'other' (if from different clades). Branch length was scaled by the maximum branch length to range from 0 to 1. The line is the least squares fit. The Pearson correlation coefficient is shown on top (C: $p \leq 2.88 \times 10^{-5}$; D: $p \leq 2.86 \times 10^{-5}$; E: $p \leq 0.018$; F: $p \leq 0.19$). (G) Module divergence scales with phylogenetic distance. Shown is a scatter plot of the relationship, for each extant (blue) and ancestral (red) species, between its phylogenetic distance to its immediate ancestor (branch length, y axis) and its AMCI (x axis). Branch length is scaled by the maximum value to range between 0 and 1. The correlation between branch length and AMCI is shown at top ($p \leq 0.033$).

Our study sheds important light on the regulatory and metabolic changes that accompanied the evolution of respiro-fermentation, which occurred twice independently in this phylogeny. In both cases, mitochondrial and respiratory genes were reassigned en masse, accompanied by convergent *cis* regulatory changes. However, changes in the regulation of amino acid, purine and sulfur metabolism genes occurred only following the WGD, possibly by repositioning of *cis*-regulatory elements relative to nucleosome free regions. The post-WGD event can be monitored in fine phylogenetic resolution, showing that many of the changes occurred gradually, with only some noticeable in *K. polysporus*, the first species to have diverged post-WGD.

Our findings raise tantalizing analogies important for human physiology. Respiro-fermentation is functionally analogous to the (somatically evolved) Warburg Effect in cancer cells (*Vander Heiden et al., 2009*), which relies—to some extent—on the 'neo'-functionalization of metabolic enzymes. There are well-known similarities in glycolysis and respiration between the two states that enable both cancer and yeast cells to direct glucose to biosynthesis, supporting their rapid growth and proliferation. However, both the causes of this metabolic signature and its connection to biosynthesis are not fully understood. The observation that genes encoding enzymes in the nucleotide salvage and glycine biosynthesis pathways are strongly induced post-diauxic shift in respiro-fermentative post-WGD species is analogous to recently identified changes in these pathways in cancer cells that display the Warburg Effect (*Gaglio et al., 2011*; *Jain et al., 2012*), suggests that rewiring of nucleotide biosynthetic pathways contributes to this phenotype and points to fundamental metabolic principles that are independent of evolutionary descent.

## Materials and methods

### An experimental framework for comparative functional genomics
#### Selection of species
The species in our panel include *S. cerevisiae* and its close relatives (*sensu stricto* clade), three other species who have diverged after the WGD (*Saccharomyces castellii*, the human pathogen *Candida glabrata*, and *Kluyveromyces polysporus*), three members of the *Kluveroymyces* clade (*Kluyveromyces waltii*, *Saccharomyces kluyveri*, and *Kluyveromyces lactis*), two members of the *Candida* clade (the human pathogen *C. albicans* and the halophile *Debaryomyces hansenii*), *Yarrowia lipolytica*, and two members of the *Schizosaccharomyces* clade (*S. pombe* and *Schizosaccharomyces japonicus*).

#### Determination of growth medium
The substantial differences in carbon lifestyle and ecological niches present major challenges in comparative functional studies. In particular, the same growth stage (e.g., mid-log) may involve different metabolic requirements and states. Indeed, not all the species grow well in typical media formulations (e.g., YPD). We first identified a growth medium ('Materials and methods') in which differences in growth between species were minimized. This formulation boosts the growth of otherwise slow growers, without substantially impacting the growth of fast growers (*Figures 1B and 2A*).

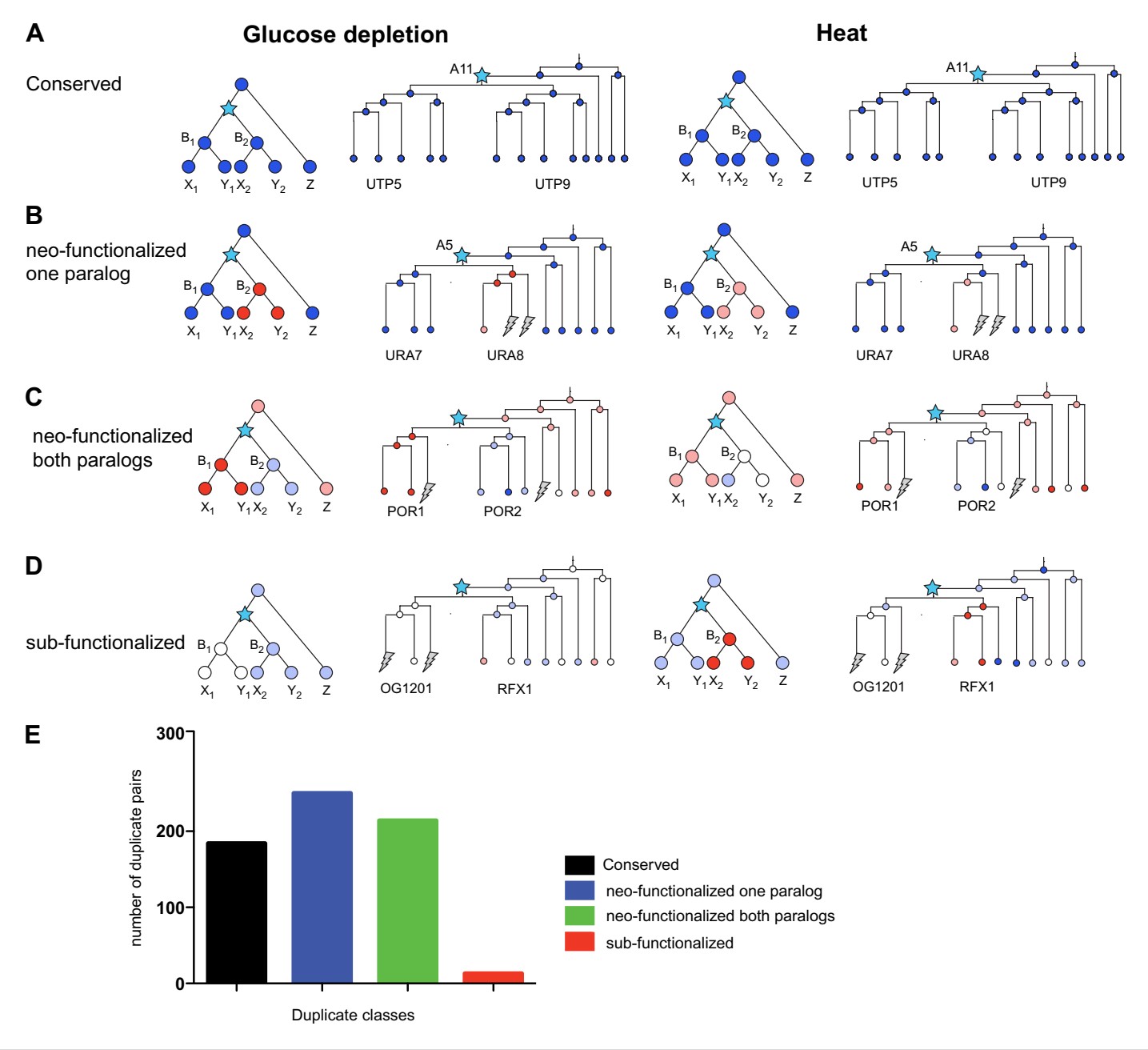

**Figure 12**. Regulatory evolution of paralogous genes in glucose depletion and heat shock. (**A**)–(**D**) several regulatory fates of paralogous genes following duplication, relative to their immediate pre-duplication ancestor in each of glucose depletion and heat shock. For each condition shown are cartoon gene trees (left) and illustrative examples from our analysis (right) representing the module assignment (circles) of each paralog and their pre-duplication ortholog in each extant and ancestral species. Module assignment is color coded as in ***Figure 3*** (Bright blue, light blue, white, pink and red from Module 1 to 5, respectively). Star: gene duplication. Lightning rod: gene loss. (**A**) Conserved: both paralogs (*UTP5* and *UTP9*) conserve the ancestral assignment (Module 1) in both responses; (**B**) Neo-functionalized, one paralog: one paralog (*URA7*) maintains the ancestral assignment (Module 1) and the other (*URA8*) is assigned to a different module (Module 5) in both responses; (**C**) Neo-functionalized, both paralogs: both paralogs (*POR1*, *POR2*) are reassigned to distinct modules than the ancestral one, but in different ways in each response. (**D**) Sub-functionalization: In glucose depletion, one paralog (*RFX1*) maintains the ancestral assignment (Module 2) and the other (*OG1201*) is reassigned (Module 3). This pattern is reversed in heat shock. (**E**) Number of paralogs pairs in each of the classes.

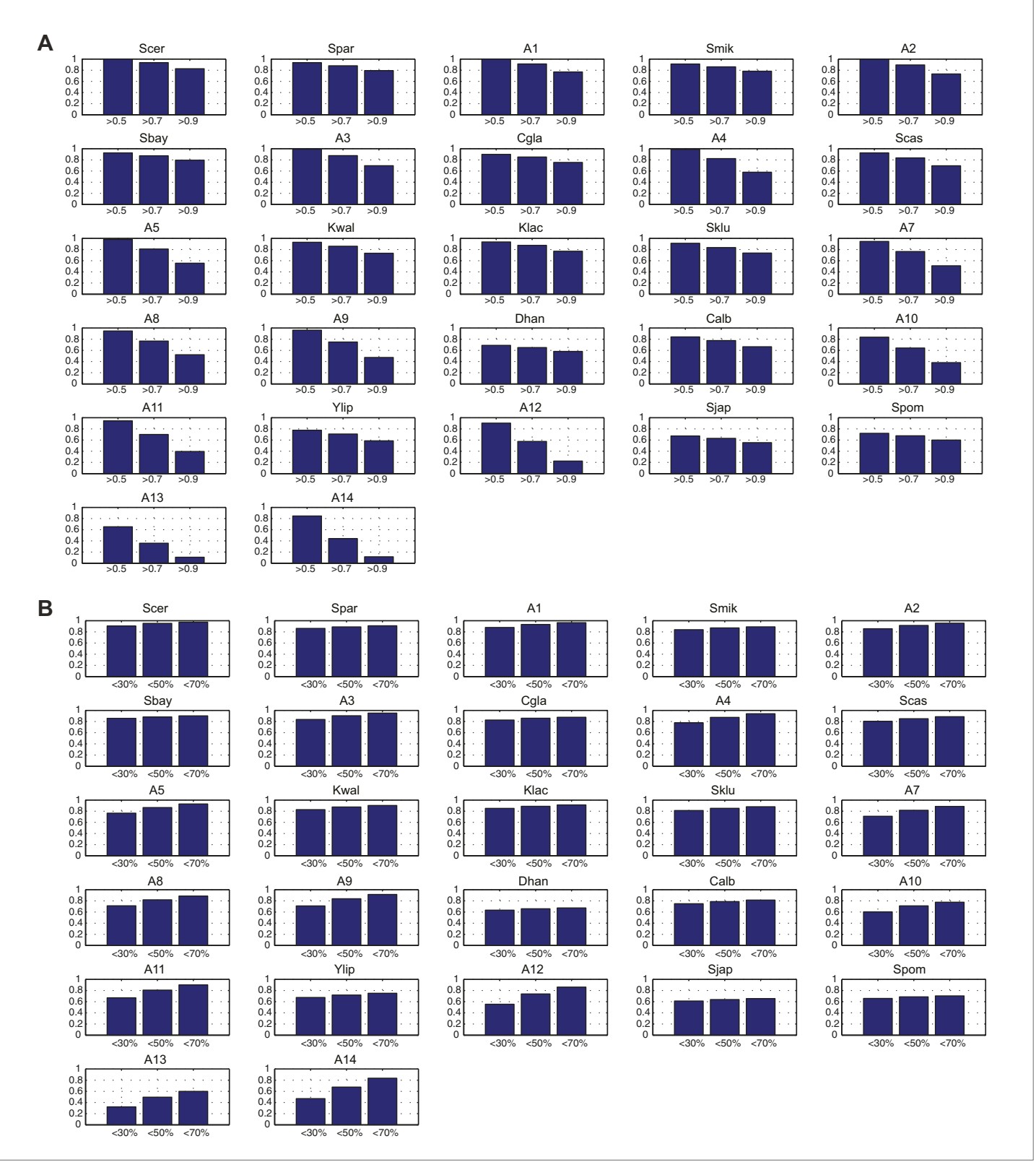

**Figure 13**. The per gene probability of Arboretum module assignments. (**A**). Shown are the fraction of genes (y axis) that are assigned to the most likely module with probability of at least 0.5, 0.7 or 0.9 in each species (x axis). (**B**). Shown are the fraction of genes (y axis) whose probabilities of the second most likely assignment is less 30%, 50%, or 70% of the most likely assignment, that is q/p<x% where q is the probability of the second most likely assignment and p is the probability of the most likely assignment.

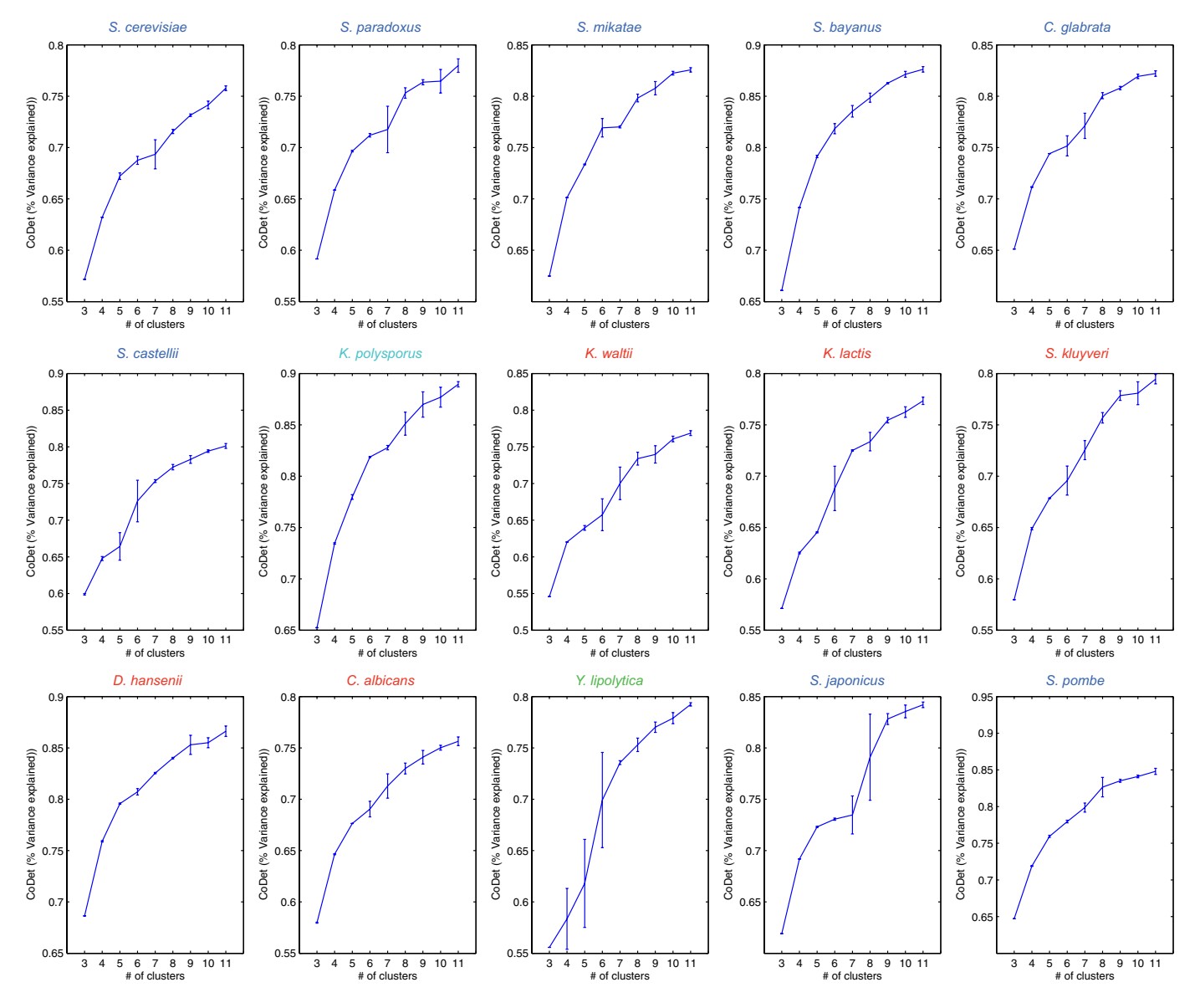

**Figure 14**. Variance captured in Arboretum modules as a function of the number of modules. Shown are the mean and standard deviation of the coefficient of determination for each species, one per plot. Mean and standard deviation were calculated for different random initializations of Arboretum runs. Coefficient of determination (y axis) was measured for different values of the number of modules (x axis).

The following figure supplements are available for figure 14:

**Figure supplement 1**. Mean expression of Aboretum modules as a function of different k values.

## Selection of physiologically comparable time points

For each species in our panel, we collected genome-wide transcriptional profiles at six physiologically-comparable time points during growth in our media, as glucose is consumed and depleted from lag phase until growth rate 'plateaus'. Even in our new medium, there is still substantial variation in growth between species, likely indicating real physiological differences (*Figure 1B*). Thus, a second challenge was to identify physiologically-comparable time points across species critical for distinguishing true inter-specific variation from temporal shifts due to growth rate differences. We therefore determined in real-time the growth rate, glucose, and ethanol levels for each species ('Materials and methods'; *Figure 2B*, *Figure 2—figure supplement 1*), and chose physiologically

comparable (but potentially physically different) time points for each species for isolating RNA from lag, mid-log, late log, 'diauxic shift' (the point at which glucose is depleted), post-shift, and plateau. The Lag phase time point was taken 30 min after inoculation for all species. Log phase was defined as the mid-point of exponential growth. Diauxic Shift is the point at which glucose levels reached 0. To choose the remaining time points, we chose two time points before the diauxic shift (Early Late Log and Late Log) and two time points after the diauxic shift (Post Shift and Late Post Shift) for each species at times proportional to the maximum growth rate in exponential phase in each species. Finally, the Plateau time point was defined as approximately 2 hr after growth had plateaued. A pilot expression study was performed using the same species-specific microarrays in which expression profiles were measured for each of the eight time points (described above) for one biological replicate for each species. These data were then used to determine the final six time points used in the larger study (*Figure 2B*, *Figure 2—figure supplement 1*).

## Quality control of microarrays

We used species specific Agilent microarrays, designed based on the ORF annotations of each species, as collected in the Fungal Orthogroup Repository (*Wapinski et al., 2007b*) (http://www.broadinstitute.org/regev/orthogroups/). We conducted each experiment in at least biological triplicates (*Figure 3—source data 1*). However, since our sampling was done in 'real time', concurrently with measuring growth rates and glucose levels, after the full time course was collected, we re-confirmed (from the growth curve, glucose curve, etc.) that the time sampled was consistent with its intended label (e.g., that diauxic shift indeed occurred at the time of glucose depletion). In a very few cases, we had to discard a planned replicate since its actual collection occurred before or after the intended physiological event (see 'Growth curve alignments' below). The collected replicates were very highly reproducible (median Pearson correlation between replicates is 0.97, *Figure 3—source data 1*). Furthermore, the microarray profiles are highly correlated to corresponding profiles measured using RNA-Seq data generated from RNA extracted from the same plateau and mid-log time point samples. This is true both when considering all genes ($0.835 < r < 0.958$, mean $r = 0.913$, 0.0362 SD) and when considering only paralogs ($0.849 < r < 0.965$, mean $r = 0.924$, 0.0368 SD), indicating that our custom microarray designs accurately distinguish paralogs and estimate their expression levels.

## Strains and growth conditions

We used the following strains for each species: *S. cerevisiae* Bb32 (3), *S. paradoxus* NCYC 2600, *S. mikatae* IFO 1815, *S. bayanus* CBS 7001, *S. bayanus uvarum* CLIB 251, *C. glabrata* CBS 138, *S. castellii* CLIB 592, *K. lactis* CLIB 209, *K. waltii* NCYC 2644, *S. kluyveri* NRRL 12651, *D. hansenii* CLIB 195, *C. albicans* SC 5314, *Y. lipolytica* CLIB 89, *S. pombe* SPY73h+ and *S. japonicus* YFS 275. All species were grown in the following rich medium (termed BMW): yeast extract (1.5%), peptone (1%), dextrose (2%), SC amino acid mix (Sunrise Science) 2 g/l, adenine 100 mg/l, tryptophan 100 mg/l, uracil 100 mg/l. The medium was chosen to minimize cross-species variation in growth. All strains were grown at 30°C except for *S. castellii*, which was grown at 25°C. *D. hansenii* is a halophilic yeast (*Kurtzman, 2000*) and was grown in BMW media made in filtered sea water purchased from PETCO.

## Media tests

We measured the growth of all our species in all published media formulations by determining the saturation coefficient (*Figure 2A*). The saturation coefficient was determined by inoculating a 3 ml culture of each media type with $1 \times 10^6$ cells/ml of each species and measuring the $OD_{600}$ using a Thermo Spectronic Genesys 20 spectrophotometer after 24 hr of growth at 30°C and 25°C. The BMW rich medium was chosen from saturation coefficient tests of over 50 formulations (made by varying each of the components listed above) as the formulation with the tightest distribution of saturation coefficients across all species (*Figure 2A*).

## Basic experimental set up

For each strain, cells were plated onto BMW plates from frozen glycerol stocks. After 2 days, cells were taken from plates and re-suspended into liquid BMW, and counted using a Cellometer Auto M10. A 3 ml BMW culture was inoculated at $1 \times 10^6$ cells/ml and placed in a New Brunswick Scientific Edison model TC-7 roller drum on the highest speed until saturated (1–2 days). The saturated cultures were

then used to inoculate 300 ml BMW batch cultures in 2-l Erlenmeyer flasks for the glucose depletion and repletion experiments described below. Flasks were transferred to New Brunswick Scientific Edison water bath model C76 shakers set to 200 rpm.

## Phenotypic characterization of each species

High-resolution growth curve data were collected for each species. Cells were grown under the conditions described in 'Basic experimental set up' above. Saturated 3 ml cultures were counted, and an appropriate amount of cells was removed from each culture to inoculate 300 ml of BMW at $1 \times 10^6$ cells/ml. The $OD_{600}$ was measured every 15–60 min using a Thermo Spectronic Genesys 20 spectrophotometer, and media samples were taken to measure glucose and ethanol levels on a YSI Biochemistry Analyzer Model 2700. These data were used to determine when each species was at a particular phase of growth to choose comparable physiological time points. The Lag phase time point was taken 30 min after inoculation for all species. Log phase was defined as the mid-point of exponential growth. Diauxic Shift is the point at which glucose levels reached 0. Two time points before the diauxic shift, Early Late Log, Late Log, and two time points after, Post Shift, Late Post Shift, were chosen for each species at a times proportional to the maximum growth rate in exponential phase in each species. Finally, the Plateau time point was defined as approximately 2 hr after the growth had plateaued). A pilot expression study was performed using the species-specific microarrays (described below) in which expression profiles were measured for each of the eight time points (described above) for one biological replicate for each species. These data were used to finalize the six time points used in the larger study (*Figure 2B*, *Figure 2—figure supplement 1*).

## Glucose depletion experiments

Cultures were grown exactly as described above. The $OD_{600}$ and glucose levels were measured throughout the day to ensure cultures were tracking with previously collected data. Samples were taken at: Lag, Log, Late log (LL), Diauxic shift (DS), Post shift (PS), and plateau (P). Samples were methanol quenched as described below and volumes removed for each sample were enough for gene expression analysis and for metabolite analysis (not shown). The actual volumes removed are shown in *Table 2*.

## Sample collection and storage

Samples were collected in 50-ml conicals filled with the appropriate amount of 100% methanol to produce a 60/40 mixture once the sample is added. The methanol-filled tubes were stored at −80°C

**Table 2.** Sample volumes for RNA extraction and metabolite analysis

| Phase | Total sample vol. (ml) | No. tubes | RNA extraction | | Metabolite analysis | | |
| | | | Vol./tube (ml) | Vol. MeOH/ tube (ml) | Vol./tube (ml) | Vol. MeOH/ tube (ml) | Vol. Water/ tube (ml) |
|---|---|---|---|---|---|---|---|
| Lag | 50 | 4 (2 Met, 2 RNA) | 12.5 | 18.75 | 12.5 | 30 | 7.5 |
| Log | 60 | 5 (3 Met, 2 RNA) | 15 | 22.5 | 10 | 30 | 10 |
| Early late log | 12 | 4 (3 Met, 1 RNA) | 6 | 9 | 2 | 9 | 4 |
| Late log | 12 | 4 (3 Met, 1 RNA) | 6 | 9 | 2 | 9 | 4 |
| Diauxic shift | 6 | 4 (3 Met, 1 RNA) | 3 | 4.5 | 1 | 9 | 5 |
| Post shift | 6 | 4 (3 Met, 1 RNA) | 3 | 4.5 | 1 | 9 | 5 |
| Late post shift | 6 | 4 (3 Met, 1 RNA) | 3 | 4.5 | 1 | 9 | 5 |
| Plateau | 6 | 4 (3 Met, 1 RNA) | 3 | 4.5 | 1 | 9 | 5 |

Shown are the appropriate culture and methanol and water volumes used in the 'cold' methanol quenching procedure for cells prior to RNA and intracellular metabolite extraction.

until ready for use. During sample collection tubes were placed in a rack in a dry-ice ethanol bath kept at approximately −40°C. Once the sample was added to the methanol, the methanol and media were separated from the cells by centrifugation and poured off. The conicals containing a cell pellet were flash frozen in liquid nitrogen and then stored at −80°C until processed for permanent storage. For permanent storage, the cell pellets were then washed in 5 ml of nuclease-free water and spun for 5 min at 3700 rpm at 4°C. The supernatant was discarded and the pellet re-suspended in 2 ml of RNAlater (Ambion, Life Technologies, Grand Island, NY) and transferred to 2 ml Sarstadt tubes for storage. The samples were left at 4°C for 24 hr before being moved to a −80°C freezer.

## RNA preparation and labeling

Total RNA was isolated using the RNeasy Midi or Mini Kits (Qiagen, Valencia, CA) according to the provided instructions for mechanical lysis. Samples were quality controlled with the RNA 6000 Nano ll kit of the Bioanalyzer 2100 (Agilent, Palo Alto, CA). Total RNA samples were labeled with either Cy3 or Cy5 using a modification of the protocol developed by Joe DeRisi (University of California at San Francisco) and Rosetta Inpharmatics that can be obtained at http://www.microarrays.org (*Wapinski et al., 2010*).

## Microarray hybridization

For each time point, either two or three biological replicates were hybridized with the Log phase sample as the reference in all cases. We used two-color Agilent 55- or 60-mer oligo-arrays in the 4 × 44 K or 8 × 15 K format for the *S. cerevisiae* strain (commercial array; four to five probes per target gene) or the custom 8 × 15 K format for all other species (two probes per target gene). After hybridization and washing per the manufacturer's instructions, arrays were scanned using an Agilent scanner and analyzed with Agilent's Feature Extraction software (release 10.5.1.). All the data has been deposited to the Gene Expression Omnibus and is available under accession GSE36253.

## Microarray data pre-processing

The median relative intensities across probes were used to estimate the expression values for each gene per replicate, and these median values across replicates were used to estimate the overall expression response per gene per time point, as previously described (*Wapinski et al., 2010*).

## Growth curve alignments

Since sampling time points were selected in real time during the experiment (based on the growth curve data collected concurrently), after the data was collected we re-confirmed that the sample time points indeed matched their expected categorization. To this end, we used two methods to align the measured growth curves. In the first method, samples from different experiments in the same species were manually aligned by overlaying growth curves for each experiment. Samples were then categorized into time point classes (LAG, LL, DS, PS, PLAT) by their position on the growth curve and their expression profiles. In the second method, we applied two transformations to align growth curves. First, in each species sampling times for growth curves of biological replicates were shifted in order to align the exponential growth phase. As expected, the doubling time for each replicate was consistent. Next, a line was fitted to the exponential growth phase using all replicate data in order to get an average growth curve. This average growth curve for each species was then aligned to the *S. cerevisiae* growth curve, adjusting for the doubling time (slope) and speed (shift along x axis) during exponential growth. Finally, we plotted dextrose depletion and used it to manually align the DS time such that it matches *S. cerevisiae*. Sampling times were then extracted from the aligned growth curve. With few exceptions, the two approaches matched and were consistent with the original sampling choice.

## Hierarchical clustering

Expression data from different species were joined into a single matrix according to their orthology relationships from gene trees generated by the Synergy algorithm (*Wapinski et al., 2007a*, *2007b*), as available from the Fungal Orthogroups Repository (http://www.broadinstitute.org/regev/orthogroups/). *K. polysporus* is not included in the repository and was added according to its orthology to *S. cerevisiae* from the Yeast Gene Order Browser (YGOB; *Byrne and Wolfe, 2005*; http://wolfe.gen.tcd.ie/ygob/). The resulting expression matrix was clustered using hierarchical clustering, weighted such that each clade is equally represented, each species within each clade is equal, and each time point within each species is

equal. The correlation coefficient between a pair of species used in the clustering procedure was calculated by using the median expression values across five time points.

## Branch length estimation

We used a tree topology and branch length estimation generated as previously described (*Wapinski et al., 2007a*, *2007b*). We used our previously published tree topology, which is based on protein sequences, except that the relative positions of two species (*C. glabrata* and *S. castellii*) have been flipped, as we previously described (*Wapinski et al., 2007b*). We estimated branch length by considering Uniform orthogroups, those where exactly one ortholog is present in every species in our panel. We randomly selected 1000 Uniform orthogroups and generated multiple sequence alignments of the genes each orthogroup using the MUSCLE program (*Edgar, 2004*). These alignments were concatenated and branch lengths were estimated using codeml of the PAML package (*Yang, 2007*). We repeated this procedure 10 times, selecting a random subset of 1000 uniform orthogroups each time, and used the average of the branch length values as branch length estimates. Average and standard deviations are reported in *Figure 1—source data 1*.

## Comparing global expression and sequence divergence across species

As a measure of expression divergence, we calculated Pearson's correlation coefficient ($r$) between the expression profiles of each pair of species at physiologically matching time points: Lag, Late lag, Diauxic shift, Post-diauxic shift and Plateau, separately (as in *Figure 3B–G*). For each pair of species, we considered only those genes that are present in both species. We used the sum of the branch lengths of each species in a pair to their nearest common ancestor as their sequence divergence, and normalized it to range from 0 to 1 by dividing by the maximum sequence distance. We next compared expression divergence and sequence divergence using linear regression.

## The Arboretum algorithm

Arboretum is an algorithm that identifies modules of co-expressed genes across multiple species and infers their evolutionary histories (*Roy et al., 2013*). Arboretum is based on a generative probabilistic model of module evolution via transition matrices (e.g., *Figure 5A*), and models expression generation at extant species via Gaussian mixture models, with a mean and variance for each time point of measurement in each species. Arboretum takes as input the genome-wide transcriptional profiles for all species measured, as well as a species tree, and all gene trees (as defined from genome sequences (*Wapinski et al., 2007a*, *2007b*). As output it provides module membership of genes in all extant and ancestral species, mean and variance of gene expression in each module in each extant species, and transition matrices at each species (extant or ancestral, except the LCA) that specify the probability of genes in each module to preserve or change their module membership in that species compared to their assignment in its immediate ancestor. During the module identification procedure, Arboretum uses the gene tree structure to trace the evolution of gene membership from ancestral modules to extant ones. If a gene changes it's module membership compared to its immediate ancestor, the ancestral module is considered to have contracted, while simultaneously the new module of the gene is considered to have expanded. Explicitly incorporating the gene trees enables Arboretum to handle complex orthology and paralogy relationships that arise from gene duplications and losses. The model parameters are learned using an Expectation Maximization (EM) algorithm, which on convergence provides maximum likelihood estimates of the transition matrices, module mean and variance and module assignments in both extant and ancestral species.

## Application of Arboretum

We applied Arboretum to two separate datasets (*Table 1*), the first (Analysis 1) including orthogroups with no duplication events (2746 orthogroups) and that are present in at least two species, one of which had to be *S. cerevisiae*, and the second (Analysis 2) including orthogroups with at most one duplication event (3676 orthogroups) and that are again present in at least one other species in addition to *S. cerevisiae*. Orthogroups with few duplications have the most reliable orthology relations and gene trees (*Wapinski et al., 2007b*). In both cases, the orthogroups could include losses, and could be lineage specific. Species and gene phylogenetic information were generated by Synergy (*Wapinski et al., 2007a*) and obtained from the Fungal Orthogroup Repository (http://www.broadinstitute.org/regev/orthogroups/) with the exception of *K. polysporus*,

which was handled as described above. *K. polysporus* was excluded from the second set (including duplicates), due to lack of reliable gene tree information for its orthogroups.

Because Arboretum is based on the EM algorithm, final module inference and parameter estimation results may depend upon initial parameter settings. Furthermore, module assignments may get permuted during learning. For these reasons, we ran Arboretum with five different random initializations, where each run was seeded with an initial clustering with merged expression data across all species. We selected a particular random initialization for our final results based on manual inspection of the transition matrices, and did not have any arbitrary module permutations. Results for different random initializations were very comparable in terms of the module expression patterns and the transition matrices. In particular, the average F-score similarity of modules between any pair of random initializations ranged from 78% to 99% for the extant species, and from 88% to 96% for ancestral species for the non-duplication case. Similarly, for the case when we included duplications, the average F-score similarity of modules between any pair of random initializations ranged from 76% to 99% for the extant species, and ranged from 77% to 98% for ancestral species. A more detailed analysis of Arboretum's stability and sensitivity is described in (*Roy et al., 2013*). The transition matrices are also very similar across random initializations for both the no-duplication (0.73–0.99) and duplication cases (0.68–0.99).

The Gaussian parameters were initialized by projecting the clusters onto individual species. The transition matrices were initialized to have diagonal-heavy elements, by setting the diagonal elements to p, p>0.5, and the off-diagonal elements to $(1 - p)/(k - 1)$ where $k$ is the number of modules. We experimented with different values of p, $p \in \{0.5, 0.6, 0.7, 0.8\}$, and used p=0.8 for the final results. Learned transition matrices are similar for p=0.6 and p=0.7, and had a correlation of (0.7–1.0), with the exception of *Schizosaccharomyces* and *C. glabrata* which exhibited a somewhat lower correlation (0.44–0.56) in the transition matrices initialized by the different p. In addition, we also compared the behavior of Arboretum using transition matrices initialized from the branch lengths. Specifically, the probability of a gene in species $s$ to maintain it's module membership was $\exp(-m \times l_s)$ where $l_s$ is the branch length from $s$ to it's immediate ancestor. We set $m$ to 1, 2 or 3, but did not observe any significant differences in the learned transition matrices using these initializations vs initializations of p=0.8. The transition matrices learned were not significantly different from those initialized uniformly with heavy diagonals (and as we observed in the case of matrices initialized with p=0.5, *C. glabrata*, and the *Schizosaccharomyces* species had the lowest correlation 0.44–0.56 with matrices initialized with p=0.8).

To select the number of modules in Arboretum we used a combination of penalized log likelihood and manual inspection. First we computed the penalized log likelihood using the minimum description length (MDL) as the penalty with different values for the number of modules, $k$ = 5, 7, 9, 11. The MDL penalty increases as a function of the number of parameters in our model, which is equal to the number of means and variances we estimate for each extant species, and the size of the transition matrices in both extant and ancestral species (see *Roy et al., 2013* for more details). Using this process the number of modules asymptotes at $k = 9$. We next inspected the patterns associated with each module, which entailed plotting the mean expression profile of each module and selecting $k$ to reflect the most distinct temporal patterns. Using this process we determined that $k = 5$ gave us the most meaningful and distinct set of profiles in terms of the GO processes enriched in different modules and the patterns associated with each module. Higher values of $k$ did not improve the GO enrichment and was prone to module switching between closely related expression profiles.

To measure the amount of signal captured by our modules, we estimated the percent variance explained, defined as $1 - \dfrac{S_{err}}{S_{total}}$, where $S_{err}$ is the sum of squared errors between the measured expression profile of a gene, and its predicted expression profile. The predicted expression profile is the mean of the module to which the gene is assigned. $S_{total}$ is defined as the total variance in the expression data. This is also called the coefficient of determination.

## Calculation of Ancestral Module Conservation Index (AMCI)

The Ancestral Module Conservation Index (AMCI) is defined for every species (except the LCA) as the tendency of the genes in that species to conserve their module assignment compared to the assignment in its immediate ancestor. The AMCI for a species $t$ is calculated as the average of the

diagonal elements of *t*'s transition matrix. Because each element is a probability value, the AMCI is bounded between 0 and 1. The closer it is to 1, the more likely is the species to preserve the module assignments of its immediate ancestor, and the closer it is to 0, the more likely it is to diverge from the module assignments of its immediate ancestor.

## Calculation of Module Contraction Index, Module Expansion Index and Module Stability

We defined two metrics for measuring module divergence: the Module Contraction Index (MCI) and the Module Expansion Index. At each phylogenetic point, *s*, we estimate for each module *m*: (1) 'innovations': the number of genes for which the module assignment is *m* in *s* but not *m* in *s*'s immediate ancestor, and (2) 'divergences': the number of genes for which the module assignment is *m* in *s*'s ancestor but not in *s*. We define the global MCI of a module as the sum of divergences across all species (extant or ancestral, except the LCA) divided by a normalization term, *Z*, defined as follows: $\sum_{s,t\in S, s\neq t} N_{st}$, where *S* is the set of all species other than the LCA, *t* is *s*'s immediate parent, and $N_{st}$ is the number of genes for which we have a module assignment in both *s* and *t*. We define MEI as the sum of all innovations divided by *Z*.

We define the stability of a module *j* by estimating the fraction of cluster assignments in a parent–child species pair that have the same assignment *j* in that pair of species.

## Enrichment analysis of Gene Ontology (GO) processes

We use the FDR corrected hyper-geometric p-value to assess enrichment of GO processes in a given gene set or time point. We use the Benjamini–Hochberg method of FDR calculation. We use the Gene Ontology terms for *S. cerevisiae* downloaded from the Saccharomyces Genome Database (SGD, http://www.yeastgenome.org/). For all other species, we use orthology to transfer the *S. cerevisiae* annotations, as previously described (*Wapinski et al., 2007b*). The GO enrichments for each Arboretum module are listed in *Supplementary files 1 and 2*.

## Enrichment analysis of *cis*-regulatory elements

We used a database of species-specific motifs (*Habib et al., 2012*) to search for *cis*-regulatory elements in 600 bp upstream of the start codon of each gene in each species. Enrichment was assessed based on the p value from the Hypergeometric distribution. The species-specific motif library was created by starting from known position weight matrices in *S. cerevisiae* (*MacIsaac et al., 2006*; *Zhu et al., 2009*) and refining them using an Expectation Maximization framework on individual species sequences. To find instances of motif matrices we used a procedure similar to *Habib et al. (2012)*. Briefly, we scan the upstream sequence of a gene summing over all instances of a motif to get a real-valued number per promoter that measures the affinity of a TF for a promoter. These numbers are ranked to select the top 40% genes with the highest affinities and these are considered as target genes for the motif.

## Contribution of gene duplication to module reassignment

We assessed the contribution of gene duplication to module reassignment at several levels. First, we considered all orthogroups with duplications as a single set. To assess the tendency of a gene to be reassigned between modules before and after duplication, we compared the normalized number of reassignments of a gene pre-duplication to the minimum, maximum and average of the normalized number of reassignments of its two 'descendant' copies post-duplication. The normalization factor for the pre-duplication counts was the number of species before duplication, and the normalization factor for the post-duplication counts was the number of species including and after the duplication. In both cases we excluded a species from the normalization if it lost the gene or had a missing value. We then used the normalized reassignment counts to generate the cumulative distributions for number of reassignments pre-duplication, and minimum, maximum and average number post-duplication, and compared these using a Kolmogorov–Smirnov (KS) test.

Second, we compared the cumulative distributions of normalized reassignment counts for genes that were never duplicated vs those that had a duplication event. For genes with duplications, we considered reassignments only at and after the point of duplication, and handled the paralogous copies as two data samples. We compared these cumulative distributions using a KS-test.

Third, we analyzed the reassignment of genes at each phylogenetic point. We considered duplicates that arose at the *Schizosaccharomyces* ancestor (A13), the *Hemiascomycota* ancestor (A11),

the *Candida* ancestor (A10), the *Kluveromycetes* and post-WGD species ancestor (A9), and the WGD ancestor (A5), because these are the phylogenetic points at which the vast majority of duplications have occurred in the considered orthogroups. For each of these phylogenetic points, $s$, we define three quantities: (1) $X$, the number of paralog pairs whose duplication occurred at $s$, (2) $Y$, the number of paralog pairs out of those in $X$ of which at least one member is reassigned at any point in the phylogeny after duplication (including $s$), and (3) $W$, the number of paralog pairs out of those in $X$ of which at least one member is reassigned at $s$ exactly. Note, each paralog pair is allowed to contribute at most once to these counts. We then estimate the fraction of paralogs that are reassigned first at their point of origin as $W/Y$, and the fraction of duplicates that are reassigned at all as $Y/X$.

Fourth, we considered duplicates that arose at each point of origin $s$, and estimated their tendency to be reassigned at $s$ and at all subsequent phylogenetic points, $t$. Specifically, we compared the number of duplicates that are reassigned at a given phylogenetic point to the total number of genes that are reassigned at that phylogenetic point. The statistical significance of the number of duplicates that are reassigned vs the number of all genes (with or without duplication) that are reassigned at that point is assessed using the Hyper-geometric distribution.

### Association of paralog pair fate with modules

As described in the main text, the module assignments of paralogs can have four possible fates right after duplication: conserved, neo-functionalized, symmetrically diverged and asymmetrically diverged. To assess if any of these classes were specifically associated with a particular expression module, we used a Hyper-geometric distribution that tests the probability that $k$ or more genes from module $i$, have fate $f$, given that there are $x$ genes in module $i$, $t$ genes in all that are associated with fate $f$, and there are a total of $T$ genes that can have any of the fates at the pre-duplication ancestral nodes. We performed this test at the pre-duplication ancestor and the post-duplication child node separately, as the total number of genes are different at these phylogenetic points.

### Nucleosome occupancy analysis

To analyze the nucleosome occupancy profiles of species-specific modules and gene sets that are reassigned in a coordinated way between modules, we used an approach similar to that previously described for functional gene sets (*Tsankov et al., 2010*). Briefly, we compared the distribution of the 5′ Nucleosome Free Region (NFR) occupancy counts for a gene set with the distribution of 5′NFR occupancy of all the genes in the genome using the KS-test. We used the negative of the logarithm of the p value of the KS-test to quantify the difference in the two distributions.

### Positioning of *cis*-regulatory motifs relative to NFRs

We used a collection of species-specific motifs (*Habib et al., 2012*) as well as motifs representing A or G rich $k$-mers (e.g., polyA and polyG) (*Tsankov et al., 2011*). Instances of these motifs were searched in 600-bp upstream of the ATG of each gene and each species. To assess if the instances of a particular motif $m$ were repositioned with respect to the NFR in post-WGD species vs pre-WGD species, we used an approach as previously described (*Tsankov et al., 2010*). For each motif $m$ and species $t$, we computed the fraction of instances in NFR regions across all of $t$'s genes, $f_{mt}$. We converted the $f_{mt}$ values to z-scores using the mean and standard deviation of all $f_{mt}$ estimated for $t$. To test if the instances of a motif $m$ are re-positioned in the post-WGD species, we compared the set of z-scores for $m$ in the post-WGD species to the set of z-scores for $m$ in the pre-WGD species, using a $t$-test ($p < 0.05$).

To assess if the instances of a motif, especially polyA, were enriched or depleted in the NFR of a query set of genes, we used a Hyper-geometric test to compare the number of instances inside the NFR vs the number of instances that are in the NFR for all genes in the genome. Similarly, we calculated enrichment of instances outside the NFR by comparing the number of genes in our query set with instances that were outside the NFR to the total number of genes that had instances outside the NFR. Depletion of instances in NFR was $(1 - p)$ of enrichment of motif instances in the NFR region. Similarly depletion outside NFR was $(1 - p)$ of enrichment of instances outside NFR.

## Acknowledgements

We thank Nir Friedman, Andrew Murray, and Tal Shay for helpful discussions and comments. We thank Leslie Gaffney for assistance with artwork.

## Additional information

### Competing interests

AR, Reviewing editor, *eLife*. The other authors declare that no competing interests exist.

### Funding

| Funder | Grant reference number | Author |
| --- | --- | --- |
| Howard Hughes Medical Institute | | Aviv Regev |
| National Institutes of Health | 2R01CA119176-01 | Aviv Regev |
| Broad Institute | | Dawn A Thompson, Aviv Regev |
| Burroughs Wellcome Fund | | Aviv Regev |
| Computing Innovation Fellow | | Sushmita Roy |
| Damon Runyon Cancer Research Foundation | | Ilan Wapinski |
| Alfred P. Sloan Foundation | | Aviv Regev |

The funders had no role in study design, data collection and interpretation, or the decision to submit the work for publication.

### Author contributions

DAT, Conception and design, Acquisition of data, Analysis and interpretation of data, Drafting or revising the article; SR, IW, Conception and design, Analysis and interpretation of data; MC, PM, JHK, Analysis and interpretation of data; MPS, CF, Conception and design, Acquisition of data, Analysis and interpretation of data; JP, AT, SN, Conception and design, Acquisition of data; AS, Acquisition of data; MK, Drafting or revising the article; AR, Conception and design, Drafting or revising the article

## Additional files

### Supplementary files

• Supplementary file 1. Gene Ontology terms enriched in Aboretum modules (analysis 1) in each species. Shown are the Gene Ontology (GO) terms that are enriched in each Aboretum module (from Analysis 1 without duplicate genes) in each ancestral and extant species. The genes that contributed to the GO enrichments are listed using the names of the *S. cerevisiae* orthologs.

• Supplementary file 2. Gene Ontology terms enriched in Aboretum modules (analysis 2) in each species. Shown are the Gene Ontology (GO) terms that are enriched in each Aboretum module (from Analysis 2 with duplicate genes) in each ancestral and extant species. The genes that contributed to the GO enrichments are listed using the names of the *S. cerevisiae* orthologs.

### Major datasets

The following datasets were generated:

| Author(s) | Year | Dataset title | Dataset ID and/or URL | Database, license, and accessibility information |
| --- | --- | --- | --- | --- |
| Thompson DA, Wapinski I, Roy S | 2012 | Evolutionary principles of modular gene regulation in Yeasts | GSE36253; http://www.ncbi.nlm.nih.gov/geo/query/acc.cgi?acc=GSE36253 | Publicly available at GEO (http://www.ncbi.nlm.nih.gov/geo/). |
| Thompson DA, Wapinski I, Roy S | 2012 | Evolutionary principles of modular gene regulation in Yeasts | GSE38478; http://www.ncbi.nlm.nih.gov/geo/query/acc.cgi?acc=GSE38478 | Publicly available at GEO (http://www.ncbi.nlm.nih.gov/geo/). |

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
