## [Decision Letter]

Thank you for submitting your work entitled “Evolutionary principles of modular gene regulation in yeasts” for further consideration at *eLife*. Your article has been favorably evaluated by a Senior editor and two reviewers, one of whom is a member of the Board of Reviewing Editors.

The paper presents an excellent and carefully analyzed dataset dealing with principles of the evolution of transcriptional regulation. The paper has already been through one round of revision and has certainly been improved. [Editors’ note: the authors were encouraged to resubmit once the manuscript describing the Arboretum algorithm had been accepted, as it serves as a basis for the analysis in the current submission.] The data look very nice, with physiology-aligned sampling points based on growth phase and nutrient availability. They apply a recently developed algorithm called Arboretum to infer co-regulated gene modules and changes in gene module membership across the phylogenetic tree. An important aspect of the approach is the ability to infer ancestral states at internal nodes across the tree, allowing comparison of module composition in extant and ancestral species.

However, both reviewers felt that the interpretation of the results based on the Arboretum algorithm needs to be revisited. This may not require additional analysis, but a more self-critical discussion seems warranted that specifies the limitations of the current approach. The major points of the referees are:

1) The Arboretum algorithm requires that “…every gene (if present in that species) must be assigned to exactly one module”. This seems to be a very crucial point for much of the analysis. One would canonically expect that most genes have pleiotropic functions, i.e., could easily be part of more than one module. We understand that it is necessary to artificially restrict this to the best-supported module, but this can at the same time lead to problems in the conclusions. For example, changes of genes from one module to another could potentially be due to small probability shifts of assignment within the algorithm. It would therefore be important to know the second best assignments for these genes and how much the assignment probabilities differ from each other. Ideally a statistical test would need to be developed that would assess whether a shift was significant. It seems that adding more analysis and discussion of this point is a prerequisite for interpreting the observed shifts between modules.

2) A major drawback is that the Arboretum approach does not exploit the interesting temporal details. Only five modules are inferred, whereas by eye it is clear from Figures 3 and 4 that there are more than five expression patterns in each species, due to subtle (but likely important) temporal differences. Simply defining modules based on “strong” versus “mild” induction or repression, without subdividing modules by temporal patterns, does not provide a lot of resolution. Furthermore, the method “over-fits” the data by classifying genes into discrete bins without saying anything quantitative about the divergence in expression. For example, genes in Figure 7 look mildly induced to similar levels in Smik and Sbay, but the middle panel shows that the genes are classified into different modules in the two species. In the end, the module classification is discrete but doesn’t really quantify the divergence in expression very well. All this leaves the biological interpretation of these modules murky – surely there must be more than five regulatory systems across the life cycle of these species, and if so then what do these module groupings really reflect?

The Discussion is currently rather short and should be expanded to deal with the above points.

---

## [Author Response]

*1) The Arboretum algorithm requires that “…every gene (if present in that species) must be assigned to exactly one module”. This seems to be a very crucial point for much of the analysis. One would canonically expect that most genes have pleiotropic functions, i.e., could easily be part of more than one module. We understand that it is necessary to artificially restrict this to the best-supported module, but this can at the same time lead to problems in the conclusions. For example, changes of genes from one module to another could potentially be due to small probability shifts of assignment within the algorithm. It would therefore be important to know the second best assignments for these genes and how much the assignment probabilities differ from each other. Ideally a statistical test would need to be developed that would assess whether a shift was significant. It seems that adding more analysis and discussion of this point is a prerequisite for interpreting the observed shifts between modules*.

We thank the reviewers for this comment. We completely agree that genes may exhibit pleiotropic behavior. In fact, Arboretum is a probabilistic method that enables us to use probabilistic module assignments, and thus a gene has a probability of belonging to a particular module and is allowed to contribute to the overall pattern of modules, with its contribution weighted by the probability. The algorithm assigns a gene to a single module *only at the very final step* of the algorithm based on the maximum probability of module assignment. Furthermore, in our set up, where we have multiple time points for a single condition our model has separate parameters (mean and variance) to model each time point.

As the reviewers point out, it is possible that there are some genes whose posterior/final probability of belonging to different modules are not different enough; that is, the difference between the probabilities associated with the most likely module and second most likely module is small. To assess how common this situation is in our data, we now examined explicitly the probabilities of our specific assignment for each gene-module-species combination (Figure 15).Author response image 1.A. Shown are the fraction of genes (y-axis) that are assigned to the most likely module with probability of at least 0.5, 0.7 or 0.9 in each species (x-axis). B. Shown are the fraction of genes (y-axis) whose probabilities of the second most likely assignment is less 30%, 50%, or 70% of the most likely assignment, that is q/p <x% where q is the probability of the second most likely assignment and p is the probability of the most likely assignment.

In principle, if there is not a high confidence of belonging to a module (or in the case of a tie), the gene’s probability would be 0.2 in all modules (assuming k=5). However, in our data, for the vast majority of the species, approximately 95% of genes are assigned to their most likely module with probability of at least 0.5 (60–70% in *C. albicans, Y. lipolytica,* and the *Schizosacchromyces).* Thus, module assignments are quite certain. Furthermore, the probabilities associated with the second most likely are less than 70% of most likely assignment for approximately 90% of genes (60–80% for *C. albicans, D. hansenii, Y. lipolytica* and the *Schizosaccharomyces* species; Figure 15). Thus, we conclude that in our particular case, using ‘hard’ module assignments at the last step is unlikely to affect our conclusions. Nevertheless, these probabilities are part of Arboretum’s output, and – as we note in our revised Discussion – future users can easily assign genes to *multiple* modules, by picking a cutoff of assigning a gene to a module, rather than picking the maximum as we did here.

*2) A major drawback is that the Arboretum approach does not exploit the interesting temporal details. Only five modules are inferred, whereas by eye it is clear from Figures 3 and 4 that there are more than five expression patterns in each species, due to subtle (but likely important) temporal differences. Simply defining modules based on “strong” versus “mild” induction or repression, without subdividing modules by temporal patterns, does not provide a lot of resolution. Furthermore, the method “over-fits” the data by classifying genes into discrete bins without saying anything quantitative about the divergence in expression. For example, genes in Figure 7 look mildly induced to similar levels in Smik and Sbay, but the middle panel shows that the genes are classified into different modules in the two species. In the end, the module classification is discrete but doesn’t really quantify the divergence in expression very well. All this leaves the biological interpretation of these modules murky – surely there must be more than five regulatory systems across the life cycle of these species, and if so then what do these module groupings really reflect*?

We thank the reviewer for this comment. This is indeed a question we were concerned with not only when we applied the algorithm, but also in its application to data. A larger number of modules would be able to distinguish finer patterns, but is also more prone to overfitting, to separating co-regulated genes into separate modules, and to inferring “transitions” between modules that are erroneous, thus over estimating divergence. In early studies with a higher number of modules, we do see separation of genes, which we know to be co-regulated by a shared mechanism (e.g., in *S. cerevisiae*), into separate modules. We therefore opted for a more conservative approach, with fewer parameters (modules) and possibly under-estimating divergence. We note this point clearly in the revised Discussion, and point the reviewers to the companion paper in *Genome Research*, where the choice of number of modules is discussed in great detail.

To address the concern of the reviewers that we are missing possible patterns, we compute the percent variance explained by our model for different values of k (i.e., different numbers of modules). We find that at k=5, 65–70% of the variance is captured. We also inspected the different patterns of expression by plotting the mean of each module. We did not find many different temporal patterns at a higher k (Figure 16).Author response image 2.Each plot is the mean expression profile of a module. Each row corresponds to different k’s and each column corresponds to a species.

A future model, explicitly handling temporal dependencies may enable us to potentially capture different dynamics, but with the current version of Arboretum higher values of k did not change the shape of the responses captured. Modeling temporal dependencies as part of the mixture model is thus an interesting future extension, as we now point out in the revised manuscript.

As for the potential over-fitting issue raised by the reviewers. First, the choice of a large number of genes in principle *reduces* the chance of splitting genes with similar expression to distinct bins, as noted above. Furthermore, Arboretum *does* provide a quantitative model of expression at the leaves (extant species) with two parameters (mean and variance of a Gaussian). Arboretum does not provide, however, a model of expression evolution (only of co-expression evolution, by evolving module assignment). Defining models of expression evolution is an important (and challenging) future goal. We point this out in the revised Discussion. Finally, we examined the expression levels of the 45 genes pointed out by the reviewers (Figure 17). We find that the mean induction for these genes is indeed lower in *S. mikatae* than in *S. bayanus,* consistent with the distinct module assignment by Arboretum, although visually these changes are hard to see in the heat map in our paper with a saturation level of −2 and +2 (but are discernible by eye when viewed at a larger range of log ratio-3 and 3; Figure 17). Most (32/45) of the genes are significantly lower in *S. mikatae* based on a paired one-sided t-test (p<0.1).Author response image 3.Shown are expression heat maps of the 45 genes for which Arboretum inferred a lower module assignment in *S. mikatae* compared to *S. bayanus.* Each heat map shows the same 45 genes at different saturation levels log ratio −1 to 1 (top), −2 to 2 (middle), −3 to 3 (bottom).

*The Discussion is currently rather short and should be expanded to deal with the above points*.

We have significantly expanded the Discussion to deal with points 1 and 2 above.